# Reinforcement Learning in Newcomblike Environments

**James Bell**
The Alan Turing Institute
London, UK
jbell@posteo.com

**Linda Linsefors**
Independent Researcher
linda.linsefors@gmail.com

**Caspar Oesterheld**
Department of Computer Science
Duke University
Durham, NC, USA
caspar.oesterheld@duke.edu

**Joar Skalse**
Department of Computer Science
University of Oxford
Oxford, UK
joar.skalse@cs.ox.ac.uk

## Abstract

Newcomblike decision problems have been studied extensively in the decision theory literature, but they have so far been largely absent in the reinforcement learning literature. In this paper we study value-based reinforcement learning algorithms in the Newcomblike setting, and answer some of the fundamental theoretical questions about the behaviour of such algorithms in these environments. We show that a value-based reinforcement learning agent cannot converge to a policy that is not *ratifiable*, i.e., does not only choose actions that are optimal given that policy. This gives us a powerful tool for reasoning about the limit behaviour of agents – for example, it lets us show that there are Newcomblike environments in which a reinforcement learning agent cannot converge to any optimal policy. We show that a ratifiable policy always exists in our setting, but that there are cases in which a reinforcement learning agent normally cannot converge to it (and hence cannot converge at all). We also prove several results about the possible limit behaviours of agents in cases where they do not converge to any policy.

## 1 Introduction

In this paper, we study decision scenarios in which outcomes depend not only on the choices made and physically implemented, but also depend directly on the agent's policy. As an example, consider an autonomous vehicle (AV) whose goal it is to arrive at target destinations quickly while minimising the probability of collisions. In practice, AVs are careful drivers. It is easy to imagine an experiment (or learning process) that might support careful driving: on each day, let the AV decide at random between a careful and a more aggressive style of driving; other drivers on the road are unaware of today's chosen driving style and therefore behave the same around the AV on both types of days. Presumably the decrease in accidents on careful days outweighs the increase in travel time.

However, imagine now that a type of AV was widely deployed. Then many of the drivers with whom the AVs interact on the road would know a lot about how these AVs behave (e.g., from reading about AVs, or from having interacted with other AVs of the same type in the past). In particular, if the other drivers know that the AVs rarely take risks, they might (whether rationally or irrationally) cut them off more, not give them right of way, etc. relative to the above experiment. Indeed, this phenomenon – human drivers bullying timid AVs – has been reported in the real world (Condliffe, 2016; Liu et al., 2020; cf. Cooper et al., 2019). As a result, the travel times of an AV are much longer if it always

follows (and is known to follow) a careful driving policy. Moreover, some of the safety benefits of careful choices disappear if the AV adopts a careful policy.

To comfortably model situations such as this one, we introduce *Newcomblike decision processes* (NDPs). The name derives from Newcomb's problem (Nozick, 1969), described in the next subsection, and similar problems that have been studied in the decision-theoretic literature. NDPs are a generalisation of Markov decision processes wherein the transition probabilities and rewards depend not only on the agent's action, but also directly on the agent's policy. Thus, for example, how aggressively other cars move depends on the timidness of the AV's policy. We describe NDPs in more detail in Sect. 1.1. Importantly, the NDP model does not assume other agents in the environment to respond rationally to the agent's policy. Thus, some NDPs cannot comfortably be modelled as games. (See Sect. 5.1 for a more detailed discussion of the relation between NDPs and game-theoretic models.)

We believe that Newcomblike dynamics are commonplace when AI systems interact with other (human or artificial) agents (Cavalcanti, 2010, Sect. 5; Oesterheld, 2019, Sect. 1; Conitzer, 2019). The deployment of AVs is rife with such dynamics. Besides aggressiveness, it might matter whether a policy is simple and thus predictable to humans, for instance. Another real-world scenario is that of recommendation systems: most readers of this paper have some idea of how these systems work and make choices based on it. ("I would like to watch this cat video, but if I do, my recommendations will soon be full of them.") Thus, the success of a particular recommendation depends not only on the recommendation itself, but also on the recommendation system's policy.

We are interested in learning to play NDPs. More specifically, we study the behaviour of value-based, model-free RL agents, who maintain a Q-function that assigns values to state–action pairs. We define these in more detail in Sect. 1.2. As we will see, such agents do not in general learn optimal policies in NDPs (as they do in MDPs). Nevertheless, we believe that studying them is an important first step in developing practical learning algorithms for NDPs due to the combination of the following points.

A) For illustrative purposes, the examples we discuss throughout this paper are simple and emphasise the dependence of the environment on the policy. However, we think that most real-world scenarios are only partially Newcomblike. For example, most of the AV's environment changes only in response to an AV's actions and does not directly depend on the AV's policy.

B) Value-based reinforcement learning algorithms are very well developed. In contrast, we would have to develop specialised learning algorithms for general NDPs from scratch.

C) As our results will show, in some situations and when set up correctly (e.g., in terms of learning rates) value-based learning converges to optimal policies, or at least to reasonable policies, even under Newcomblike dynamics. For example, in a game of rock-paper-scissors against an opponent who knows the agent's policy, some value-based learning agents learn the optimal policy of mixing uniformly.

In light of A–C, we think that the most realistic paths to developing learning algorithms for real-world scenarios with Newcomblike dynamics will involve value-based RL. Specifically, one avenue toward realistic algorithms is to develop extensions of value-based RL and can detect and correct failures that might arise in Newcomblike dynamics. For that, we need to understand how value-based RL behaves in NDPs.

**Contributions** In Sect. 2 we demonstrate that value-based RL algorithms can only converge to a policy that is *ratifiable* – that is, to a policy $\pi$ for which all actions taken by $\pi$ have optimal expected reward when following $\pi$. In Sect. 3, we discuss the convergence properties of agents in Newcomblike situations, and show that there are cases where value-based agents must fail to converge. The action frequencies might converge, even when the policies do not. In Sect. 4, we establish some conditions on any action frequency that an agent could converge to. We also show that there are decision problems and agents where even the action frequencies do not converge.

### 1.1 Newcomblike Decision Processes

A *Newcomblike decision process (NDP)* is a tuple $\langle S, A, T, R, \gamma \rangle$ where $S$ is a finite set of *states*; $A$ is a finite set of *actions*; $T\colon S \times A \times (S \rightsquigarrow A) \rightsquigarrow S$ is a nondeterministic *transition function*; $R\colon S \times A \times S \times (S \rightsquigarrow A) \rightsquigarrow \mathbb{R}$ is a nondeterministic *reward function*, which we assume to be bounded; and $\gamma \in [0, 1)$ is a *discount factor*.

A policy $\pi : S \rightsquigarrow A$ is a function that nondeterministically maps states to actions. We use $\pi(a \mid s)$ to denote the probability of taking action $a$ in state $s$ while following the policy $\pi$. $T$ and $R$ are functions from states, actions, and policies. In other words, they allow the outcome of a decision to depend on the distributions from which the agent draws its actions, rather than just the state and the action that is in fact taken. Also note that $T(s, a, \pi)$ and $R(s, a, s', \pi)$ are defined even if $\pi(a \mid s) = 0$. We say that an NDP is a *bandit NDP* if it has only one state. We will sometimes use $R(s, a, \pi)$ as a shorthand for $R(s, a, T(s, a, \pi), \pi)$, and we will sometimes omit the state from $T$, $R$, and $\pi$ for bandit NDPs. Moreover, we normally let $\gamma = 0$ for bandit NDPs.

Consider a distribution over initial states for an agent, and let $\pi$ be its policy, let $x_t$ be the sequence of states it visits and $a_t$ the sequence of actions it takes. We say $\pi$ is optimal for that distribution if it maximises $\mathbb{E}[\sum_{i=0}^{\infty} \gamma^i R(x_i, a_i, a_{i+1}, \pi)]$. Note that unlike in the MDP case the optimal policy does depend on the initial distribution, however this isn't relevant in the bandit case.

As an example consider the eponymous Newcomb's Problem.

**Newcomb's Problem** (Nozick, 1969): There are two boxes in front of you; one opaque box, and one transparent box. You see that the transparent box contains \$1,000. You can choose to either take only the opaque box, or to take both boxes. The boxes have been placed in this room by an agent who can predict your policy; if he believes that you will take only the opaque box then he has put \$1,000,000 in the opaque box, but if he believes that you will take both boxes then he has left the opaque box empty. Do you take one box, or two?

A version of Newcomb's Problem can be formalised as the following bandit NDP: $S = \{s\}$, $A = \{a_1, a_2\}$,

$$R(a_1, \pi) = \begin{cases} 0 & \text{w.p.} & \pi(a_2) \\ 10 & \text{w.p.} & \pi(a_1) \end{cases} \quad \text{and } R(a_2, \pi) = \begin{cases} 5 & \text{w.p.} & \pi(a_2) \\ 15 & \text{w.p.} & \pi(a_1) \end{cases},$$

where "w.p." is short for "with probability". The key feature of this NDP is that, for any fixed policy, $a_2$ ("two-boxing") yields a higher reward than $a_1$ ("one-boxing"). But the expected reward of a policy increases in $\pi(a_1)$ s.t. the optimal policy is to always play $a_1$.[1] We can view Newcomb's problem as a simple version of the AV dynamic described in the introduction, where $a_2$ is a driving action that allows other drivers to cut the AV off at no risk.

We say that an NDP is *continuous* if $T$ and $R$ are continuous in the policy. In this paper we work mainly with continuous NDPs. This is in part because it is technically convenient, and in part because we believe that continuity is satisfied in many realistic cases.[2]

## 1.2 Reinforcement Learning Agents

We consider value-based reinforcement learning agents. Such agents have two main components; a *Q-function* $S \times A \rightarrow \mathbb{R}$ that predicts the expected future discounted reward conditional on taking a particular action in a particular state, and a *bandit algorithm* that is used to select actions in each state based on the $Q$-function. Given a policy $\pi$, we use $q_\pi(a \mid s)$ to denote the (true) expected future discounted reward conditional on taking action $a$ in state $s$ while following the policy $\pi$ (and conditional on all subsequent actions being chosen by $\pi$). A model-free agent will update $Q$ over time to make it converge to $q_\pi$ when following $\pi$. If $Q$ is represented as a lookup table, the agent is said to be *tabular*. If the state space is large, it is common to instead approximate $q_\pi$ (with e.g. a neural network). For simplicity, we focus mostly on tabular agents. However, some of our results (Theorems 2 and 5) only assume that $Q$ converges to $q_\pi$ (for some $q_\pi$) and therefore apply immediately to non-tabular agents, as long as the function approximator for $q_\pi$ converges to the same $q_\pi$.

---

[1] In most versions of Newcomb's Problem, the predictor directly predicts the agent's action with some fixed accuracy, and the agent is unable to randomise in a way that is unpredictable to the environment. This version of the problem can be modelled as a regular MDP. However, we believe that our version is more realistic in the context of AI. After all, AIs can at least act pseudo-randomly, while the distribution according to which they choose is predictable if e.g. their source code is known.

[2] For example, even if the environment has direct access to the source code of the agent, it may in general not be feasible to extract the exact action probabilities from the code. However, it is always possible to estimate the action probabilities by sampling. If this is done then $T$ and $R$ will depend continuously on the policy.

The $Q$-values can be updated in different ways. One method is to use the update rule

$$Q_{t+1}(a_t \mid s_t) \leftarrow (1 - \alpha_t(s_t, a_t))\, Q_t(a_t \mid s_t) + \alpha_t(s_t, a_t)(r_t + \gamma \max_a Q_t(a \mid s_{t+1})),$$

where $a_t$ is the action taken at time $t$, $s_t$ is the state visited at time $t$, $r_t$ is the reward obtained at time $t$, and $\alpha_t(s, a)$ is a learning rate. This update rule is known as *Q-learning* (Watkins, 1986). Other widely used update rules include *SARSA* (Rummery and Niranjan, 1994) and *Expected SARSA* (van Seijen et al., 2009). For the purposes of this paper it will not matter significantly how the $Q$-values are computed, as long as it is the case that if an agent converges to a policy $\pi$ in some NDP and explores infinitely often then $Q$ converges to $q_\pi$. We will later see that this is the case for $Q$-learning, SARSA, and Expected SARSA in continuous NDPs.

There are also several different bandit algorithms. Two types of agents that are widely used in practice and that we will refer to throughout the paper are *softmax agents* and *$\epsilon$-Greedy agents*. The policy of a softmax agent with a sequence of temperatures $\beta_t \in \mathbb{R}_+$ is given by:

$$\pi_t(a \mid s) = \frac{\exp(Q_t(a \mid s)/\beta_t)}{\sum_{a' \in A} \exp(Q_t(a' \mid s)/\beta_t)}.$$

Unless otherwise stated we assume that $\beta_t \to 0$. The policy of an $\epsilon$-Greedy agent with a sequence of exploration probabilities $\epsilon_t \in [0, 1]$ is $\pi_t(a \mid s) = 1 - \epsilon_t$ if $a = \arg\max_{a'} Q_t(a' \mid s)$ and $\pi_t(a \mid s) = \epsilon_t/(|A| - 1)$ otherwise. Unless otherwise stated we assume that $\epsilon_t \to 0$. We assume that $\epsilon$-Greedy breaks ties for argmax, so that there is always some $a \in A$ such that $\pi(a \mid s) = 1 - \epsilon_t$. We say that an agent is *greedy in the limit* if the probability that the agent takes an action that maximises $Q$ converges to 1, and we say that it *explores infinitely often* if it takes every action in every state infinitely many times.

## 1.3   Some Initial Observations

We here make three simple observations about NDPs that we will use to prove and understand the results throughout this paper. First, a continuous NDP always has, for each possible distribution over initial states, a policy $\pi$ that maximises the expected discounted reward $\mathbb{E}[R \mid \pi]$, since $\mathbb{E}[R \mid \pi]$ exists and is continuous in $\pi$, and since the set of possible policies is a compact set. Also note that an NDP in which $T$ or $R$ is discontinuous may not have any such policy.

Second, whereas all MDPs have a deterministic optimal policy, in some NDPs all optimal policies randomise. To see this we introduce another example we will look at in this paper.

**Death in Damascus** (Gibbard and Harper, 1976): Death will come for you tomorrow. You can choose to stay in Damascus (where you are currently) or you can flee to Aleppo. If you are in the same city as Death tomorrow, you will die. Death has already decided which city he will go to – however, he can predict your policy, and has decided to go to the city where he believes that you will be tomorrow. Do you stay in Damascus, or flee to Aleppo?

We formalise this as the bandit NDP $S = \{s\}$, $A = \{a_{\text{Damascus}}, a_{\text{Aleppo}}\}$, and

$$R(a_{\text{Damascus}}, \pi) = \begin{cases} 0 & \text{w.p.} & \pi(a_{\text{Damascus}}) \\ 10 & \text{w.p.} & \pi(a_{\text{Aleppo}}) \end{cases} \quad \text{and} \quad R(a_{\text{Aleppo}}, \pi) = \begin{cases} 10 & \text{w.p.} & \pi(a_{\text{Damascus}}) \\ 0 & \text{w.p.} & \pi(a_{\text{Aleppo}}) \end{cases},$$

where "w.p." is again short for "with probability". In this NDP, randomising uniformly between $a_{\text{Damascus}}$ and $a_{\text{Aleppo}}$ is the unique optimal policy and in particular outperforms both deterministic policies.

Note also that the Bellman optimality equation does not hold for NDPs. Even in Newcomb's Problem, as described above, Bellman's optimality equation is not satisfied by the optimal policy.

## 2   Ratifiability

If an agent in the limit only takes the actions with the highest $Q$-values and it converges to some policy $\pi_\infty$, then it is clear that, for a given state, all actions in the support of $\pi_\infty$ must have equal expected utility given $\pi_\infty$. Otherwise, the $Q$-values would eventually reflect the differences in expected utility and the agent would move away from $\pi_\infty$. Similarly, if the algorithm explores sufficiently often, the

actions that are taken with limit probability $0$ cannot be better given $\pi_\infty$ than those taken by $\pi_\infty$. After all, if they were better, the agent would have eventually figured this out and assigned them large probability.

This condition on $\pi_\infty$ resembles a well-known doctrine in philosophical decision theory: ratificationism (see Weirich, 2016, Sect. 3.6, for an overview). One form of ratificationism is based on a distinction between a *decision* – what the agent chooses – and the *act* that is selected by that decision. Very roughly, ratificationism then states that a decision is rational only if the acts it selects have the highest expected utility given the decision. Concepts of causality are often invoked to formalise the difference between the decision, the act, and their respective consequences. Our setup, however, has such a differentiation built in: we will view the policy as the "decision" and the action sampled from it as the "act".

## 2.1 Strong Ratifiability

As hinted earlier, slightly different versions of the concept of ratifiability are relevant depending on how much exploration a learning algorithm guarantees. We start with the stronger version, which more closely resembles what decision theorists mean when they speak about ratifiability.

**Definition 1.** *Let $M \subseteq S$ be a set of states. A policy $\pi$ is strongly ratifiable on $M$ if $\mathrm{supp}(\pi(\cdot \mid s)) \subseteq \arg\max_{a \in A} q_\pi(a \mid s)$ for all $s \in M$.*

In Newcomb's Problem the only strongly ratifiable policy is to play $a_2$ with probability $1$. In Death in Damascus, only the optimal policy (mixing uniformly) is strongly ratifiable. There can also be several strongly ratifiable policies. For example, if you play the Coordination Game of Table 1 against an opponent who samples his action from the same policy as you then there are three strongly ratifiable policies; to select action $a$ with probability $1$, to select action $b$ with probability $1$, and to select $a$ with probability $1/3$ and $b$ with probability $2/3$.

**Theorem 2.** *Let $\mathcal{A}$ be a model-free reinforcement learning agent, and let $\pi_t$ and $Q_t$ be $\mathcal{A}$'s policy and $Q$-function at time $t$. Let $\mathcal{A}$ satisfy the following in a given NDP:*

- *$\mathcal{A}$ is greedy in the limit, i.e. for all $\delta > 0$, $\mathbb{P}\left(Q_t(\pi_t(s)) \leq \max_a Q_t(a \mid s) - \delta\right) \to 0$ as $t \to \infty$.*
- *$\mathcal{A}$'s $Q$-values are accurate in the limit, i.e. if $\pi_t \to \pi_\infty$ as $t \to \infty$, then $Q_t \to q_{\pi_\infty}$ as $t \to \infty$.*

*Then if $\mathcal{A}$'s policy converges to $\pi_\infty$ then $\pi_\infty$ is strongly ratifiable on the states that are visited infinitely many times.*

In Appendix A we show that the $Q$-values of a tabular agent are accurate in the limit in any continuous NDP if the agent updates its $Q$-values with SARSA, Expected SARSA, or $Q$-learning, given that the agent explores infinitely often and uses appropriate learning rates. Since we would expect most well-designed agents to have accurate $Q$-values in the limit, Theorem 2 should apply very

|   | $a$ | $b$ |
|---|-----|-----|
| $a$ | 2,2 | 0,0 |
| $b$ | 0,0 | 1,1 |

Table 1: The Coordination Game

broadly. Using Kakutani's fixed-point theorem, it can be shown that every continuous NDP has a ratifiable policy.

**Theorem 3.** *Every continuous NDP has a strongly ratifiable policy.*

Of course, the fact that a ratifiable policy always exists does not necessarily mean that a reinforcement learning agent must converge to it – we will consider the question of whether or not this is the case in Sect. 3. It is also worth noting that a discontinuous NDP may not have any strongly ratifiable policy.

It is a topic of ongoing discussion among philosophical decision theorists whether (strong) ratifiability should be considered a normative principle of rationality, see Weirich (2016, Sect. 3.6) for details. In general, the policy $\pi$ that maximises $\mathbb{E}[R \mid \pi]$ may or may not be ratifiable, as shown by Death in Damascus and Newcomb's problem, respectively.

There is a correspondence between ratificationism and many game-theoretic concepts. For example, if you are playing a zero-sum game against an opponent who can see your policy and plays some distribution over best responses to it then $\pi$ can only be ratifiable if it is a maximin strategy. To give another example, if you are playing a symmetric game against an opponent who follows the same policy as you then $\pi$ is ratifiable if and only if $(\pi, \pi)$ is a Nash equilibrium. Joyce and Gibbard (1998, Sect. 5) discuss the relation in more detail.

## 2.2 Weak Ratifiability

We now show that even without infinite exploration, $\pi_\infty$ must still satisfy a weaker notion of ratifiability.

**Definition 4.** *Let $M \subseteq S$ be a set of states. A policy $\pi$ is weakly ratifiable on $M$ if $q_\pi(a \mid s)$ is constant across $a \in \mathrm{supp}(\pi(s))$ for all $s \in M$.*

What makes this a weak version of ratifiability is that it does not put any requirements on the expected utility of actions that $\pi$ does not take, it merely says that all actions that $\pi$ takes with positive probability must have the same (actual) $q$-value. As a special case, this means that all deterministic policies are weakly ratifiable. This includes one-boxing in Newcomb's problem. Nonetheless, there are bandit NDPs in which the optimal policy is not even weakly ratifiable. For example, consider an NDP with actions $a_1$, $a_2$, where $R(a_1, \pi) = -100(\pi(a_1) - 1/2)^2 + 1$ and $R(a_2, \pi) = -100(\pi(a_1) - 1/2)^2$. The optimal policy mixes close to uniformly ($\pi(a_1) = 101/200$), but this is not weakly ratifiable, because $R(a_1, \pi) > R(a_2, \pi)$.

**Theorem 5.** *Same conditions as Theorem 2, but where $\mathcal{A}$'s Q-values are only required to be accurate in the limit for state-action pairs that $\mathcal{A}$ visits infinitely many times. Then $\pi_\infty$ is weakly ratifiable on the set of states that are visited infinitely many times.*

# 3 Non-Convergence of Policies

We have shown that most reinforcement learning algorithms can only converge to (strongly) ratifiable policies. We now consider the question of whether they always converge to a policy at all. We find that this is not the case.

## 3.1 Theoretical Results

From Theorem 2 it follows that in e.g. Death in Damascus an $\epsilon$-Greedy agent who explores infinitely often cannot converge to any policy. After all, the only strongly ratifiable policy (and thus limit policy) is to mix uniformly and an $\epsilon$-Greedy agent never mixes uniformly.

Perhaps more surprisingly, there are also NDPs in which a (slow-cooling) softmax agent cannot converge to any policy. As an example, consider a bandit NDP with three actions $a_1, a_2, a_3$, and where the rewards $R(a_i, \pi)$ have expectations

$$\pi(a_{i+1}) + 4 \cdot 13^3 \cdot \pi(a_i) \mathbb{1}\left[\forall j : \pi(a_j) \geq 1/4\right] \prod_j \left(\pi(a_j) - 1/4\right). \tag{1}$$

For $i = 3$, we here let $a_{i+1} = a_1$. We also require that the rewards are stochastic with a finite set of outcomes such that the empirical $Q$-values are never exactly equal between different actions. We call this the *Repellor Problem*. It has only one strongly ratifiable policy (mixing uniformly), but – as illustrated by Figure 1 – when the current policy mixes close to uniformly, the softmax agent learns (in expectation) to play less uniformly.

**Theorem 6.** *Let $\mathcal{A}$ be an agent that plays the Repellor Problem, explores infinitely often, and updates its Q-values with a learning rate $\alpha_t$ that is constant across actions, and let $\pi_t$ and $Q_t$ be $\mathcal{A}$'s policy and Q-function at time $t$. Assume also that for $j \neq i$, if $\pi_t(a_i)$, $\pi_t(a_j)$ both converge to positive values, then*

$$\frac{\pi_t(a_i) - \pi_t(a_j)}{Q_t(a_i) - Q_t(a_j)} \xrightarrow[a.s.]{} \infty \tag{2}$$

*as $t \to \infty$. Then $\pi_t$ almost surely does not converge.*

Line 2 is satisfied, for example, for softmax agents with $\beta_t$ converging to 0. Recall also that e.g. $Q$-learning and SARSA are equivalent for bandit NDPs (if $\gamma = 0$).

## 3.2 Empirical Results

Empirically, softmax agents converge (to strongly ratifiable policies) in many NDPs, provided that the temperature decreases sufficiently slowly. To illustrate this we will use *Aymmetric Death in Damascus*, a version of Death in Damascus wherein the rewards of $a_{\text{Aleppo}}$ are changed to be 5 (instead of 0) with probability $\pi(a_{\text{Aleppo}})$ and (as before) 10 with the remaining probability. This NDP has only one (strongly) ratifiable policy, namely to go to Aleppo with probability $2/3$ and Damascus with probability $1/3$. This is also the optimal policy. We use this asymmetric version to make it easier to distinguish between convergence to the ratifiable policy and the default of uniform mixing at high temperatures. Figure 2 shows the probability of converging to this policy with a softmax agent and a plot of the policy on one run. We can see that this agent reliably converges provided that the cooling is sufficiently slow.

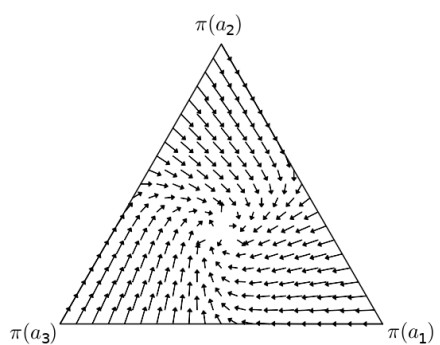

Figure 1: The triangle shows the space of possible policies in the Repellor Problem, parameterised by the probability they assign to each of the three actions. Plotted against this space is the expected direction in which a softmax agent would change its policy if playing a particular policy.

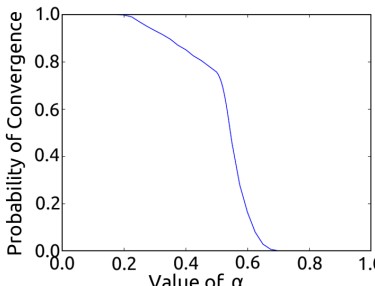 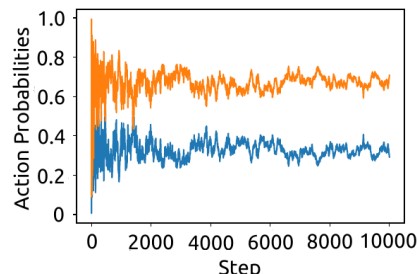

Figure 2: The left figure plots the probability of softmax converging in Asymmetric Death in Damascus given $\beta_n = n^{-\alpha}$ against $\alpha$. More accurately it is a plot of the fraction of runs which assigned a $Q$-value of at least 5.5 to the action of going to Aleppo after 5000 iterations. These are empirical probabilities from 20,000 runs for every $\alpha$ that is a multiple of 0.025, and 510,000 runs for each $\alpha$ that is a multiple of 0.005 between 0.5 and 0.55. Notice the "kink" at $\alpha = 0.5$. Based on our experiments, this kink is not an artefact and shows up reliably in this kind of graph. The right-hand figure shows how the action probabilities evolve over time for a single run (chosen to converge to the mixed strategy) for $\alpha = 0.3$.

However, there are also fairly simple games in which it seems like softmax agents cannot converge. Consider *Loss-Averse Rock-Paper-Scissors* (LARPS), the problem of playing Rock-Paper-Scissors against an opponent that selects each action with the same probability as you, and where you assign utility 1 to a win, 0 to a draw, and -10 to a loss. We conjecture that slow-cooling softmax agents do not converge in LARPS. We have unfortunately not been able to prove this formally, but Figure 3 presents some empirical data which corroborates the hypothesis.

## 4   Convergence of Action Frequencies

We have seen that there are NDPs in which some reinforcement learning algorithms cannot converge to any policy. But if they do not converge to any policy, what does their limit behaviour look like? We now examine whether these algorithms converge to taking each action with some limit *frequency*, and what sorts of frequencies they can converge to.

## 4.1 Possible Frequencies in the Bandit Case

In this section we establish a number of conditions that must be satisfied by any limit action frequency of a value-based agent. We consider agents that converge to deterministic policies (such as $\epsilon$-Greedy agents), and we limit our analysis to the bandit case (with $\gamma = 0$).

Let $P_t^{\Sigma} \colon A \to [0, 1]$ be the frequency with which each action in $A$ is taken in the first $t$ steps (for some agent and some bandit NDP). Note that $P_t^{\Sigma}$ is a random variable. By the law of large numbers, $P_t^{\Sigma}(a) - {}^{1}/{t} \sum_{i=0}^{t} \pi_i(a)$ converges to 0 almost surely as $t \to \infty$. Let $\pi_a$ be the policy that takes action $a$ with probability 1, and let $q_a = q_{\pi_a}$.

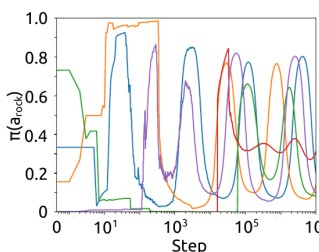

Figure 3: This figure shows five runs of a softmax agent in LARPS, and plots $\pi(a_{\text{rock}})$ against the total number of episodes played. The agent's $Q$-values are the historical mean rewards for each action, and $\beta_t = 1/\log t$.

**Theorem 7.** *Assume that there is some sequence of random variables $(\epsilon_t \geq 0)_t$ s.t. $\epsilon_t \xrightarrow[t\to\infty \ a.s.]{} 0$ and for all $t \in \mathbb{N}$ it is*

$$\sum_{a^* \in \arg\max_a Q_t(a)} \pi_t(a^*) \geq 1 - \epsilon_t. \tag{3}$$

*Let $P_t^{\Sigma} \to p^{\Sigma}$ with positive probability as $t \to \infty$. Then across all actions $a \in \text{supp}(p^{\Sigma})$, $q_a(a)$ is constant.*

That is, the actions played with positive limit frequency must all be equally good when played deterministically. This condition is vaguely analogous to weak ratifiability, and is proven in roughly the same way as Theorem 2.

**Theorem 8.** *Same assumptions as Theorem 7. If $|\text{supp}(p^{\Sigma})| > 1$ then for all $a \in \text{supp}(p^{\Sigma})$ there exists $a' \in A$ s.t. $q_a(a') \geq q_a(a)$.*

This condition is an instability condition. Say that multiple actions are taken with nonzero limit frequency, and that action $a$ has the highest $Q$-value at time $t$. Then for other actions to be played with positive limit frequency, other actions must at some point be believed to be optimal again (since the probability of exploration goes to zero). Hence they cannot all be worse when explored while mainly playing $a$, since $a$ could otherwise be played forever.

**Theorem 9.** *Same assumptions as Theorem 7. Let $U$ be the $Q$-value $q_a(a)$ which (by Theorem 7) is constant across $a \in \text{supp}(p^{\Sigma})$. For any $a' \in A - \text{supp}(p^{\Sigma})$ that is played infinitely often, let frequency 1 of the exploratory plays of $a'$ happen when playing a policy near elements of $\{\pi_a \mid a \in \text{supp}(p^{\Sigma})\}$. Then either there exists $a \in \text{supp}(p^{\Sigma})$ such that $q_a(a') \leq U$; or $q_{a'}(a') < U$.*

Theorem 9 describes what circumstances are needed for an actions $a'$ to be played with limit frequency zero. One possibility is that exploration is done only finitely many times (in which case bad luck could lead to low $Q$-values). A second possibility is that the exploration mechanism is "rigged" so that $a'$ is mostly played when playing policies outside the proximity of $\{\pi_a \mid a \in \text{supp}(p^{\Sigma})\}$. In this case the utility of $a'$ under some zero-limit-frequency policy might lead to low $Q$-values. If exploration of $a'$ is spread out more naturally then all but frequency zero of that exploration will happen near elements of $\{\pi_a \mid a \in \text{supp}(p^{\Sigma})\}$. In this case, the only reason for $a'$ to be played with zero frequency is that exploring $a'$ near some of the elements of $\{\pi_a \mid a \in \text{supp}(p^{\Sigma})\}$ makes $a'$ look poor.

## 4.2 When is Frequency Convergence Possible?

We believe there are NDPs in which an $\epsilon$-Greedy agent cannot converge to any limit action frequency. Specifically, we believe that LARPS is such an example. Figure 4a shows the directions in which the frequencies of different actions evolve. The graph seems to have no attractor and hence we believe an $\epsilon$-Greedy agent cannot converge to any limit action frequency in this NDP. We have not been able to rigorously prove this. However, experiments seem to confirm this hypothesis. Figure 4b depicts five

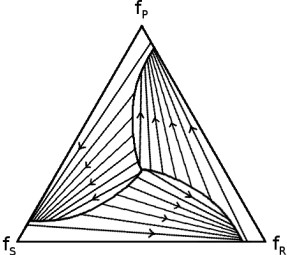

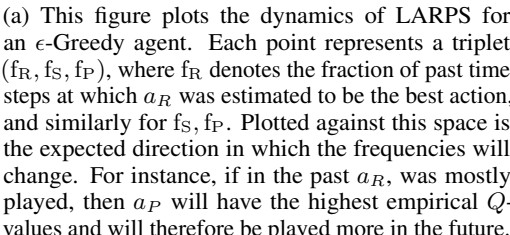

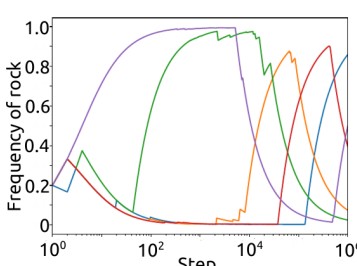

(a) This figure plots the dynamics of LARPS for an $\epsilon$-Greedy agent. Each point represents a triplet $(f_R, f_S, f_P)$, where $f_R$ denotes the fraction of past time steps at which $a_R$ was estimated to be the best action, and similarly for $f_S, f_P$. Plotted against this space is the expected direction in which the frequencies will change. For instance, if in the past $a_R$, was mostly played, then $a_P$ will have the highest empirical $Q$-values and will therefore be played more in the future.

(b) This figure shows five runs of an $\epsilon$-Greedy agent in LARPS, and plots the proportion of past episodes in which the agent played "rock" against the total number of episodes played. The agent's $Q$-values are the historical mean rewards for each action, and its $\epsilon$-value is 0.01.

Figure 4

runs of $\epsilon$-Greedy in LARPS. We can see that the agents oscillate between different actions, and that the periods increase in length.

## 5 Related Work

### 5.1 Learning in games

Some of the Newcomblike dynamics we have described in this paper could also be modelled as games, especially as so-called *Stackelberg games* in which one player, the *Stackelberg leader*, chooses a strategy first and another player, the *Stackelberg follower*, then responds to that strategy. For example, in the case of autonomous vehicles (AVs), we might imagine that the AV company is the Stackelberg leader and the human drivers are the Stackelberg followers.

That said, there are differences between NDPs and games. NDPs can react arbitrarily to the agent's policy, whereas in games, the other players play a best (i.e., expected-utility-maximising) response. In many real-world situations, other agents in the environment cannot be comfortably modelled as expected-utility-maximising agents. Interactions between AVs and humans can serve as examples. Most people probably do not reason rationally about small-probability, big-impact events, such as car crashes. Also, humans will generally operate on simplified models of an AV's policy (even when more detailed models are available). Of course, a game-theoretic analysis can also be fruitful and address issues that we ignore: By assuming all players to be rational, game theory can provide recommendations and predictions for multiple agents simultaneously, while our NDP model considers a single agent for a given environment. We believe that the NDP perspective provides additional insight into learning in such situations.

Despite the differences between NDPs and games, there are some interesting parallels between model-free learning in NDPs and in games, where similar learning methods are sometimes referred to as *fictitious play* (Brown, 1951). Fudenberg and Levine (1998, Chapter 2) show that fictitious play can only converge to a Nash equilibrium (for similar results about convergence to Nash equilibrium, see e.g. Mazumdar et al., 2020, Oesterheld et al., 2021). As noted in Sect. 2.1, the concept of Nash equilibrium resembles the concept of ratifiability. Shapley (1964) shows that fictitious play can fail to converge. However, there are many special cases in which convergence is guaranteed, including two-player zero-sum games (Robinson, 1951) and generic $2 \times 2$ games (Miyasawa, 1961).

### 5.2 Learning and Newcomblike problems

Other authors have discussed learning in Newcomblike problems. The most common setup is one in which the learner assigns values directly to policies, or more generally to that which the agent chooses.

It is then usually shown that (among the policies considered) the agent will converge to taking the one with the highest (so-called evidential) expected utility (Albert and Heiner, 2001; Oesterheld, 2018). This contrasts with our setup, in which the learner selects policies but assigns values to actions. Oesterheld (2019) also studies agents who maximise reward in Newcomblike environments. However, Oesterheld does not consider the learning process. Instead he assumes that the agent has already formed beliefs and uses some form of expected utility maximisation. He also specifically considers the implications of having some overseer assign rewards based on beliefs about the state of the world (as opposed to having the reward come directly from the true world state).

# 6   Discussion and Further Work

We have seen that value-based reinforcement learning algorithms can fail to converge to any policy in some NDPs, and that when they do converge, they can only converge to *ratifiable* policies. Decision theorists have discussed whether ratifiability should be considered to be a sound normative principle. Note that (as discussed in Sect. 2) the optimal policies $\pi$ are not in general ratifiable. We have also examined the limit action *frequencies* that agents can converge to (even when the policies do not converge). Still, there are NDPs in which many agents cannot converge even to any such frequency. We gave some results on what actions can be taken with positive limit frequency. A loose connection to ratifiability can still be drawn.

Overall, established decision-theoretical ideas can be used to understand and formally describe the behaviour of "out-of-the-box" reinforcement learning agents in NDPs. However, their behaviour is not always desirable. Our work elucidates possible failures. We hope that our work will thus enable more accurate reasoning about the behaviour of RL agents in real-world situations, especially when interacting with humans or other agents. We hold such improvements in understanding to be broadly beneficial to the safe design and deployment of AI systems.

Throughout the paper, we have noted specific open questions related to our results. For instance, can the results in Sect. 4.1 be generalised beyond the bandit setting? There are also many topics and questions about our setting that we have not touched on at all. For instance, our experimental results indicate that convergence is often slow (considering how simple the given problems are). It might be desirable to back up this impression with theoretical results. We have only studied simple value-based model-free algorithms – the analysis could be extended to other reinforcement learning algorithms (e.g., policy-gradient or model-based algorithms). Also, there are further ways in which we could generalise our setting. One example is to introduce partial observability and imperfect memory into the NDPs. The latter has been studied in game and decision theory (Piccione and Rubinstein, 1997; Elga, 2000), but recently – under the name *memoryless POMDP* – also in reinforcement learning (Azizzadenesheli et al., 2016; Steckelmacher et al., 2018; cf. Conitzer, 2019). What makes this especially appealing in the NDP context is that problems related to imperfect memory relate closely to Newcomblike problems (Briggs, 2010; Schwarz, 2015). One could also look for narrower classes of NDPs in which RL agents are guaranteed to perform well in some sense.

Ultimately, the goal of this line of research is to develop learners that are able to deal effectively and safely with Newcomblike dynamics. We hope that our results will be useful in developing extensions of value-based RL that can detect and correct for the failures that arise when existing methods are applied in Newcomblike settings. However, we should also consider alternative approaches that do not hinge on insights from the present work. For example, a few recent papers (on learning in non-Newcomblike settings) have considered learning to predict the expected utility as a function of the *policy* (as opposed to traditional $Q$ values, which are not paremeterised by the policy) (Harb et al., 2020). In principle, learning such a *policy evaluation function* avoids the problems of the learners considered in this paper. However, it remains to be seen how practical this approach is.

## Acknowledgements

We thank our anonymous reviewers for their exceedingly helpful comments. We thank the Foundational Research Institute (now the Center on Long-Term Risk) and the funders and organisers of the AI Safety Camp 2018 on Gran Canaria for supporting this project. Caspar Oesterheld is thankful for support by the National Science Foundation under Award IIS-1814056.

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
