# A   *Q*-value convergence

We here show that if a tabular agent converges to a policy $\pi_\infty$ in a continuous NDP then $Q_t$ converges to $q_{\pi_\infty}$, assuming that the agent updates its $Q$-values in an appropriate way. To prove this we will use the following lemma:

**Lemma 10.** *Let $\langle \zeta_t, \delta_t, F_t \rangle$ be a stochastic process where $\zeta_t, \delta_t, F_t : X \to \mathbb{R}$ satisfy*

$$\delta_{t+1}(x) = (1 - \zeta_t(x_t)) \cdot \delta_t(x_t) + \zeta_t(x_t) \cdot F_t(x_t)$$

*with $x_t \in X$ and $t \in \mathbb{N}$. Let $P_t$ be a sequence of increasing $\sigma$-fields such that $\zeta_0$ and $\delta_0$ are $P_0$-measurable and $\zeta_t$, $\delta_t$ and $F_{t-1}$ are $P_t$-measurable, $t \geq 1$. Then $\delta_t$ converges to $0$ with probability $1$ if the following conditions hold:*

1. *$X$ is finite.*

2. *$\zeta_t(x_t) \in [0, 1]$ and $\forall x \neq x_t : \zeta_t(x) = 0$.*

3. *$\sum_t \zeta_t(x_t) = \infty$ and $\sum_t \zeta_t(x_t)^2 < \infty$ with probability 1.*

4. *$\mathrm{Var}\{F_t(x_t) \mid P_t\} \leq K(1 + \kappa \|\delta_t\|_\infty)^2$ for some $K \in \mathbb{R}$ and $\kappa \in [0, 1)$.*

5. *$\|\mathbb{E}\{F_t \mid P_t\}\|_\infty \leq \kappa \|\delta_t\|_\infty + c_t$, where $c_t \to 0$ with probability 1 as $t \to \infty$.*

*where $\|.\|_\infty$ is a (potentially weighted) maximum norm.*

*Proof.* See Singh et al. (2000). ∎

We say that a $Q$-value update rule is *appropriate* if it has the following form;

$$Q_{t+1}(a_t \mid s_t) \leftarrow (1 - \alpha_t(a_t, s_t)) \cdot Q_t(a_t \mid s_t) + \alpha_t(a_t, s_t) \cdot (r_t + \gamma \cdot \hat{v}_{t+1}(s_{t+1})),$$

where $\hat{v}_t(s)$ is an estimate of the value of $s$, and if moreover

$$\lim_{t \to \infty} \mathbb{E}\left[\hat{v}_t(s) - \max_a Q_t(a \mid s)\right] = 0.$$

$Q$-learning is of course appropriate. Moreover, SARSA and Expected SARSA are also both appropriate, if the agent is greedy in the limit. Note that since $R$ is bounded, $Q_t(a \mid s)$ has bounded support. This means that if for all $\delta > 0$, $\mathbb{P}(Q_t(\pi_t(s) \mid s) \leq \max_a Q_t(a \mid s) - \delta) \to 0$ as $t \to \infty$, then $\mathbb{E}_{a \sim \pi_t}[Q_t(a \mid s)] \to \max_a Q_t(a \mid s)$ as $t \to \infty$.

**Theorem 11.** *In any continuous NDP $\langle S, A, T, R, \gamma \rangle$, if a tabular agent converges to a policy $\pi_\infty$ then $Q_t$ converges to $q_{\pi_\infty}$, if the following conditions hold:*

1. *The agent updates its $Q$-values with an appropriate update rule.*

2. *The update rates $\alpha_t(a, s)$ are in $[0, 1)$, and for all $s \in S$ and $a \in A$ we have that $\sum_t \alpha_t(a, s) = \infty$ and $\sum_t \alpha_t(a, s)^2 < \infty$ with probability 1.*

Note that condition 2 requires that the agent takes every action in every state infinitely many times

*Proof.* Let

- $X = S \times A$

- $\zeta_t(a, s) = \alpha_t(a, s)$

- $\delta_t(a, s) = Q_t(a \mid s) - q_{\pi_\infty}(a \mid s)$

- $F_t(a, s) = r_t + \gamma \hat{v}_{t+1}(s_{t+1}) - q_{\pi_\infty}(a \mid s)$

Since $S$ and $A$ are finite, and since $R$ is bounded, we have that condition 1 and 4 in Lemma 10 are satisfied. Moreover, assumption 2 of this theorem corresponds to condition 2 and 3 in Lemma 10. It remains to show that condition 5 is satisfied, which we can do algebraically:

$$\|\mathbb{E}\{F_t \mid P_t\}\|_\infty$$

$$= \max_{s,a} \left| \mathbb{E}\Big[ r_t + \gamma \hat{v}_t(s_{t+1}) - q_{\pi_\infty}(a \mid s) \Big] \right|$$

$$= \max_{s,a} \left| \mathbb{E}\Big[ r_t + \gamma \max_{a'} Q_t(a' \mid s_{t+1}) - q_{\pi_\infty}(a \mid s) + \gamma \hat{v}_t(s_{t+1}) - \gamma \max_{a'} Q_t(a' \mid s_{t+1}) \Big] \right|$$

$$\leq \max_{s,a} \left| \mathbb{E}\Big[ r_t + \gamma \max_{a'} Q_t(a' \mid s_{t+1}) - q_{\pi_\infty}(a \mid s) \Big] \right| + \max_{s,a} \left| \mathbb{E}\Big[ \gamma \hat{v}_t(s_{t+1}) - \gamma \max_{a'} Q_t(a' \mid s_{t+1}) \Big] \right|$$

Note that the second term in this expression is bounded above by

$$\max_s \left| \mathbb{E}\Big[ \hat{v}_t(s) - \max_a Q_t(a \mid s) \Big] \right|$$

Let us use $k_t$ to denote this expression. Since the $Q$-value update rule is appropriate we have that $k_t \to 0$ as $t \to \infty$. We thus have:

$$= \max_{s,a} \left| \mathbb{E}[r_t + \gamma \max_{a'} Q_t(a' \mid s_{t+1}) - q_{\pi_\infty}(a \mid s)] \right| + k_t$$

We can now expand the expectations, and rearrange the terms:

$$\begin{aligned}
= \max_{s,a} \Bigg| &\sum_{s' \in S} \mathbb{P}(T(s, a, \pi_t) = s') \\
&\Big( \mathbb{E}[R(s, a, s', \pi_t)] + \gamma \max_{a'} Q_t(a' \mid s') \Big) \\
- &\sum_{s' \in S} \mathbb{P}(T(s, a, \pi_\infty) = s') \\
&\Big( \mathbb{E}[R(s, a, s', \pi_\infty)] + \gamma \max_{a'} q_{\pi_\infty}(a' \mid s') \Big) \Bigg| + k_t
\end{aligned}$$

$$\begin{aligned}
= \max_{s,a} \Bigg| &\sum_{s' \in S} \mathbb{P}(T(s, a, \pi_\infty) = s') \\
&\Big( \mathbb{E}[R(s, a, s', \pi_t)] + \gamma \max_{a'} Q_t(a' \mid s') - \\
&\quad \mathbb{E}[R(s, a, s', \pi_\infty)] - \gamma \max_{a'} q_{\pi_\infty}(a' \mid s') \Big) \\
+ &\sum_{s' \in S} \Big( \mathbb{P}(T(s, a, \pi_t) = s') - \\
&\quad \mathbb{P}(T(s, a, \pi_\infty) = s') \Big) \cdot X \Bigg| + k_t
\end{aligned}$$

where $X = \mathbb{E}[R(s, a, s', \pi_t)] + \gamma \max_{a'} Q_t(a' \mid s')$. Let $d_t(s, a)$ be the second term in this expression, and let $b_t(s, a, s') = \mathbb{E}[R(s, a, s', \pi_t)] - \mathbb{E}[R(s, a, s', \pi_\infty)]$. Since $\pi_t \to \pi_\infty$, and since $T$ and $R$ are continuous, we have that $b_t(s, a, s') \to 0$ and $d_t(s, a) \to 0$ as $t \to \infty$ (for any $s$, $a$, and $s'$). We

thus have:

$$= \max_{s,a} \left| \sum_{s' \in S} \mathbb{P}(T(s, a, \pi_\infty) = s') \right.$$

$$\left( \gamma \max_{a'} Q_t(a' \mid s') - \gamma \max_{a'} q_{\pi_\infty}(a' \mid s') + \right.$$

$$\left. \left. b_t(s, a, s') \right) + d_t(s, a) \right| + k_t$$

$$\leq \gamma \max_{s,a} \left| Q_t(a \mid s) - q_{\pi_\infty}(a \mid s) \right| +$$

$$\max_{s,a,s'} \left| b_t(s, a, s') + d_t(s, a) + k_t \right|$$

$$= \gamma \max_{s,a} \left| \delta(s, a) \right| + c_t = \gamma \|\delta_t\|_\infty + c_t$$

where $c_t = \max_{s,a,s'} \left| b_t(s, a, s') + d_t(s, a) + k_t \right|$. This means that

$$\|\mathbb{E}\{F_t \mid P_t\}\|_\infty \leq \gamma \|\delta_t\|_\infty + c_t$$

where $\gamma \in [0, 1)$ and $c_t \to 0$ as $t \to \infty$. Thus by lemma 10 we have that $Q_t$ converges to $q_{\pi_\infty}$.  □

## B  Proof of Theorem 2

**Theorem 2.** *Let $\mathcal{A}$ be a model-free reinforcement learning agent, and let $\pi_t$ and $Q_t$ be $\mathcal{A}$'s policy and Q-function at time $t$. Let $\mathcal{A}$ satisfy the following in a given NDP:*

- *$\mathcal{A}$ is greedy in the limit, i.e. for all $\delta > 0$, $\mathbb{P}\left(Q_t(\pi_t(s)) \leq \max_a Q_t(a \mid s) - \delta\right) \to 0$ as $t \to \infty$.*
- *$\mathcal{A}$'s Q-values are accurate in the limit, i.e. if $\pi_t \to \pi_\infty$ as $t \to \infty$, then $Q_t \to q_{\pi_\infty}$ as $t \to \infty$.*

*Then if $\mathcal{A}$'s policy converges to $\pi_\infty$ then $\pi_\infty$ is strongly ratifiable on the states that are visited infinitely many times.*

*Proof.* Let $\pi_t \to \pi_\infty$ and hence $Q_t \to q_{\pi_\infty}$. For strong ratifiability, we have to show that for all actions $a'$ and states $s$, if $a'$ is suboptimal (in terms of true $q$ values) given $\pi_\infty$ in $s$, then $\pi_\infty(a' \mid s) = 0$.

If $a'$ is suboptimal in this way, then there is $\delta > 0$ s.t.

$$q_{\pi_\infty}(a' \mid s) \leq \max_a q_{\pi_\infty}(a \mid s) - \delta.$$

Thus, since $Q_t \to q_{\pi_\infty}$, it is for large enough $t$,

$$Q_t(a' \mid s) \leq \max_a Q_t(a \mid s) - \frac{\delta}{2}.$$

By the greedy-in-the-limit condition, $\pi_t(a' \mid s) \to 0$. Because $\pi_t \to \pi_\infty$, it follows that $\pi_\infty(a' \mid s) = 0$, as claimed.  □

## C  Proof of Theorem 3

**Lemma 12** (Kakutani's Fixed-Point Theorem). *Let $X$ be a non-empty, compact, and convex subset of some Euclidean space $\mathbb{R}^n$, and let $\phi : X \to 2^X$ be a set-valued function s.t. $\phi$ has a closed graph and s.t. $\phi(x)$ is non-empty and convex for all $x \in X$. Then $\phi$ has a fixed point.*

*Proof.* See Kakutani (1941).  □

**Theorem 3.** *Every continuous NDP has a strongly ratifiable policy.*

*Proof.* Let $N = \langle S, A, T_N, R_N, \gamma \rangle$ be a continuous NDP, and let $N_\pi$ be the MDP $\langle S, A, T_{N_\pi}, R_{N_\pi}, \gamma \rangle$ that is obtained by fixing the dynamics in $N$ according to $\pi$ – that is, $T_{N_\pi}(s, a) = T_N(s, a, \pi)$, and $R_{N_\pi}(s, a, s') = R_N(s, a, s', \pi)$. Let $\phi_N : (S \rightsquigarrow A) \to 2^{(S \rightsquigarrow A)}$ be the set-valued function s.t. $\phi_N(\pi)$ is the set of all policies that are optimal in $N_\pi$. We will show that the graph of $\phi_N$ is closed and apply Kakutani's fixed point theorem.

Suppose $(\pi_i)$ is a sequence of policies converging to $\pi_0$ and suppose $\lambda_i \in \phi_N(\pi_i)$ is a sequence converging to $\lambda_0$. For all sufficiently large $i$, $\text{supp}(\lambda_0) \subseteq \text{supp}(\lambda_i)$ (as the state and action spaces are finite). Therefore for sufficiently large $i$, $\lambda_0 \in \phi_N(\pi_i)$. By the continuity with respect to $\pi$ of $\mathbb{E}[R \mid \lambda_0]$ in $N_\pi$, $\lambda_0 \in \phi_N(\pi_0)$. Therefore, the graph of $\phi_N$ is closed.

The domain of $\phi_N$ is a non-empty, compact, convex subset of Euclidean space. Any MDP always has an optimal policy, and so $\phi_N(\cdot)$ is non-empty. Since $N_\pi$ is an MDP $\phi_N(\pi)$ is a set of deterministic policies and all their convex combinations, and so $\phi_N(\cdot)$ is convex. Hence, by Kakutani's Fixed Point Theorem, there must be a $\pi$ s.t. $\pi \in \phi_N(\pi)$. Then $\pi$ is strongly ratifiable in $N$. Hence every continuous NDP has a strongly ratifiable policy. $\square$

## D   Proof of Theorem 6

To prove Theorem 6, we first need to prove the following lemma.

**Lemma 13.** *Let $X_t$ be a non-negative discrete stochastic process, indexed by $t$, and let $\mathcal{F}_t$ denote the history upto time $t$. Suppose $X_t$ is bounded, i.e. there exists $B$ such that $X_t \leq B$, and further that $|X_{t+1} - X_t| < B/t$. Suppose also that there exists $\epsilon > 0$ and $b > 0$ such that whenever $X_t < b$,*

$$Var(X_{t+1}|\mathcal{F}_t) \geq \frac{\epsilon}{t^2} \tag{4}$$

*and*

$$\mathbb{E}[X_{t+1}|\mathcal{F}_t] - X_t \geq 0. \tag{5}$$

*Then $\mathbb{P}(X_t \to 0) = 0$.*

*Proof.* Let $a_n = 2^{2^n}$ and define the following sequences of events. Firstly, letting $s_n$ denote $2^n \sqrt{4B^2 \sum_{t=a_{n+1}}^{\infty} \frac{1}{t^2}}$,

$$A_n = \left\{ X_{a_{n+1}} > s_n \right\} \tag{6}$$

and

$$A'_n = A_n \vee \left\{ \exists t \in [a_n, a_{n+1}] \text{ s.t. } X_t \geq b \right\}, \tag{7}$$

which tell us that at some point after time $a_n$, but not after $a_{n+1}$, the value of $X_t$ isn't very small and secondly

$$B_n = \{X_t < b \forall t \geq a_n\}. \tag{8}$$

This event is useful because it is implied by convergence to $0$ and tells us that Equation 5 can be applied.

We will show that two properties hold. Firstly that $\mathbb{P}(A'_n \wedge B_n \wedge \{X_t \to 0\}) \leq 2^{-2n}$ and secondly that $\mathbb{P}(A'_n | \mathcal{F}_{a_n}) \geq 2/5$ for all sufficiently large $n$.

From the second of these properties, and the fact that $A'_n$ is $\mathcal{F}_{a_{n+1}}$ measurable, it is immediate by the argument of the Borel-Cantelli Lemma that, almost surely, $A'_n$ occurs infinitely often (i.o.) i.e. for infinitely many $n$. From this and the fact that $X_t \to 0 \implies (B_n \forall n$ sufficiently large) we can deduce the following

$$\mathbb{P}(X_t \to 0) \tag{9}$$
$$= \mathbb{P}(B_n \wedge \{X_t \to 0\} \forall n \text{ sufficiently large}) \tag{10}$$
$$= \mathbb{P}((A'_n \wedge B_n \wedge \{X_t \to 0\}) \text{ i.o.}) \tag{11}$$
$$\leq \mathbb{P}(\exists n > m \text{ s.t. } A'_n \wedge B_n \wedge \{X_t \to 0\}) \tag{12}$$
$$\leq \sum_{n=m}^{\infty} \mathbb{P}(A'_n \wedge B_n \wedge \{X_t \to 0\}). \tag{13}$$

It is immediate from the first fact that this sum is convergent, and thus it must converge to zero as $m \to \infty$, but $m$ was arbitrary so $\mathbb{P}(X_t \to 0) = 0$.

We now prove the first property. Note that if $B_n$ occurs then $A'_n$ can only occur if $A_n$ occurs. Thus $\mathbb{P}(A'_n \wedge B_n \wedge \{X_t \to 0\}) \leq \mathbb{P}(B_n \wedge \{X_t \to 0\}|A_n)$. To see this is small, we consider an augmentation of $X_t$ given by

$$Y_t = \begin{cases} X_t & t \leq a_{n+1} \\ Y_{t-1} + (X_t - X_{t-1}) \\ \quad - \mathbb{E}[X_t - X_{t-1}] & t > a_{n+1}. \end{cases} \tag{14}$$

Note that this process is a martingale (for $t > a_{n+1}$), i.e. $\mathbb{E}[Y_{t+1}|\mathcal{F}_t] = Y_t$ for all $t > a_{n+1}$, and that if $B_n$ occurs then $Y_t \leq X_t$ for all $t$ (by Equation 5). As $Y$ is a martingale $\mathbb{E}[Y_t|\mathcal{F}_{a_{n+1}}] = Y_{a_{n+1}}$. Furthermore we can compute as follows

$$\text{Var}(Y_t|\mathcal{F}_{a_{n+1}}) \tag{15}$$

$$= \mathbb{E}[(Y_t - Y_{a_{n+1}})^2|\mathcal{F}_{a_{n+1}}] \tag{16}$$

$$= \mathbb{E}[(\sum_{r=a_{n+1}}^{t-1} Y_{r+1} - Y_r)^2|\mathcal{F}_{a_{n+1}}] \tag{17}$$

$$= \mathbb{E}[\sum_{r=a_{n+1}}^{t-1} \sum_{s=a_{n+1}}^{t-1} (Y_{r+1} - Y_r)(Y_{s+1} - Y_s)|\mathcal{F}_{a_{n+1}}] \tag{18}$$

$$= \sum_{r=a_{n+1}}^{t-1} \sum_{s=a_{n+1}}^{t-1} \mathbb{E}[(Y_{r+1} - Y_r)(Y_{s+1} - Y_s)|\mathcal{F}_{a_{n+1}}]. \tag{19}$$

As $Y$ is a martingale we have that this final expectation is zero unless $r = s$. To see this assume WLOG that $r > s$ and note that

$$\mathbb{E}[(Y_{r+1} - Y_r)(Y_{s+1} - Y_s)|\mathcal{F}_{a_{n+1}}] \tag{20}$$

$$= \mathbb{E}[\mathbb{E}[(Y_{r+1} - Y_r)(Y_{s+1} - Y_s)|\mathcal{F}_r]|\mathcal{F}_{a_{n+1}}] \tag{21}$$

$$= \mathbb{E}[\mathbb{E}[(Y_{r+1} - Y_r)|\mathcal{F}_r)(Y_{s+1} - Y_s)|\mathcal{F}_{a_{n+1}}] \tag{22}$$

$$= \mathbb{E}[0(Y_{s+1} - Y_s)|\mathcal{F}_{a_{n+1}}] \tag{23}$$

$$= 0. \tag{24}$$

Putting these together, along with the fact that $Y_{r+1} - Y_r \leq 2B/r$ (which follows from the similar bound on difference in $X$), we get that

$$\text{Var}(Y_t|\mathcal{F}_{a_{n+1}}) = \sum_{r=a_{n+1}}^{t-1} \mathbb{E}[(Y_{r+1} - Y_r)^2|\mathcal{F}_{a_{n+1}}] \tag{25}$$

$$\leq 4B^2 \sum_{r=a_{n+1}}^{\infty} r^{-2}. \tag{26}$$

Thus, for all $t \geq a_{n+1}$, by Chebyshev's inequality,

$$\mathbb{P}(Y_t < 0|A_n) \leq \mathbb{P}(|Y_t - Y_{a_{n+1}}| > Y_{a_{n+1}}|A_n) \tag{27}$$

$$\leq \mathbb{P}\left(|Y_t - Y_{a_{n+1}}| > s_n|A_n\right) \tag{28}$$

$$\leq \frac{\text{Var}(Y_t|\mathcal{F}_{a_{n+1}})}{s_n^2} \tag{29}$$

$$\leq 2^{-2n}. \tag{30}$$

Whilst by the final property if $B_n$ occurs and $X_t \to 0$ then $Y_t < \eta$ for all sufficiently large $t$ for all $\eta > 0$. Thus $\mathbb{P}(B_n \wedge \{X_t \to 0\}|A_n) \leq 2^{-2n}$ and $\mathbb{P}(A'_n \wedge B_n \wedge \{X_t \to 0\}) \leq 2^{-2n}$.

We now prove that $\mathbb{P}(A'_{n+1}|\mathcal{F}_{a_{n+1}}) \geq 2/5$ for sufficiently large $n$, where we have replaced $n$ by $n+1$ for convenience. We again define $Y_t$ exactly as for the previous property and note again that

it is a martingale and that, for $t \geq a_{n+1}$, $4B^2/t^2 \geq \mathrm{Var}(Y_{t+1}|\mathcal{F}_t) \geq \epsilon/t^2$. Thus we can apply the martingale central limit theorem (Hall and Heyde, 1980, Theorem 5.4) to conclude that, setting $\sigma_n^2 = \mathrm{Var}(Y_{a_{n+1}} - Y_{a_n}|\mathcal{F}_{a_n})$, the distribution conditioned on $\mathcal{F}_{a_{n+1}}$ of $(Y_{a_{n+2}} - Y_{a_{n+1}})/\sigma_{n+1}$ converges to a standard normal distribution as $n \to \infty$. Let $Z$ have a standard normal distribution.

$$
\begin{aligned}
\mathbb{P}(Y_{a_{n+2}} > s_{n+1}) &= \mathbb{P}((Y_{a_{n+2}} - Y_{a_{n+1}})/\sigma_{n+1} > (s_{n+1} - Y_{a_{n+1}})/\sigma_{n+1}) \\
&= \mathbb{P}((Y_{a_{n+2}} - Y_{a_{n+1}})/\sigma_{n+1} > (s_{n+1} - X_{a_{n+1}})/\sigma_{n+1}) \\
&\geq \mathbb{P}((Y_{a_{n+2}} - Y_{a_{n+1}})/\sigma_{n+1} > s_{n+1}/\sigma_{n+1}) \\
&\to \mathbb{P}(Z > \lim_{n \to \infty} s_{n+1}/\sigma_{n+1}) \\
&= \mathbb{P}(Z > 0) = \frac{1}{2}
\end{aligned}
$$

Where the limit in the probability was zero because $s_{n+1} = O(2^{n+1-3\cdot 2^{n+1}})$ and $\sigma_{n+1} = \Omega(2^{-3\cdot 2^n})$. Finally note that, $X_t \geq Y_t$ for all $t \leq a_{n+2}$ unless the event $\{\exists a_{n+1} \leq t \leq a_{n+2} \text{s.t.} X_t \geq b\}$ occurs. So for sufficiently large $n$ either $\{\exists a_{n+1} \leq t \leq a_{n+2} \text{s.t.} X_t \geq b\}$ or, with probability at least $2/5$, $A_{n+1}$ occurs. Therefore, for sufficiently large $n$, $\mathbb{P}(A'_{n+1}|\mathcal{F}_{a_{n+1}}) \geq 2/5$ and the proof is complete. $\qquad\square$

**Theorem 6.** *Let $\mathcal{A}$ be an agent that plays the Repellor Problem, explores infinitely often, and updates its Q-values with a learning rate $\alpha_t$ that is constant across actions, and let $\pi_t$ and $Q_t$ be $\mathcal{A}$'s policy and Q-function at time $t$. Assume also that for $j \neq i$, if $\pi_t(a_i)$, $\pi_t(a_j)$ both converge to positive values, then*

$$
\frac{\pi_t(a_i) - \pi_t(a_j)}{Q_t(a_i) - Q_t(a_j)} \underset{a.s.}{\to} \infty \tag{2}
$$

*as $t \to \infty$. Then $\pi_t$ almost surely does not converge.*

*Proof.* We first need to establish the fact that $(1/3, 1/3, 1/3)$ is the only strongly ratifiable policy. First, if $\pi(a_j) \leq 1/4$ for some $j$ then $\mathbb{E}[R(a_i, \pi)] = \pi(a_{i+1})$. It is easy to see that for this reward function, there is no strongly ratifiable policy other than the symmetric $(1/3, 1/3, 1/3)$.

The other case of $\pi(a_j) > 1/4$ for all $j$ is harder. Finding strongly ratifiable policies in this range gives rise to the following system of polynomial equations, constrained to $p_1, p_2, p_3 \in [1/4, 1]$:

$$
\begin{aligned}
p_1 + 4 \cdot 13^3 p_2 \left(p_1 - \frac{1}{4}\right)\left(p_2 - \frac{1}{4}\right)\left(p_3 - \frac{1}{4}\right) &= x \\
p_2 + 4 \cdot 13^3 p_3 \left(p_1 - \frac{1}{4}\right)\left(p_2 - \frac{1}{4}\right)\left(p_3 - \frac{1}{4}\right) &= x \\
p_3 + 4 \cdot 13^3 p_1 \left(p_1 - \frac{1}{4}\right)\left(p_2 - \frac{1}{4}\right)\left(p_3 - \frac{1}{4}\right) &= x \\
p_1 + p_2 + p_3 &= 1
\end{aligned}
$$

Although this is non-trivial, it can be solved by computer algebra system.[3] For completeness, we would like to give a more human argument here. Consider the simpler system

$$
p_1 + Kp_2 = p_2 + Kp_3 = p_3 + Kp_1 \tag{31}
$$
$$
p_1 + p_2 + p_3 = 1 \tag{32}
$$

Note that for $p_1, p_2, p_3$ to satisfy the original system of equations, it has to satisfy the above system of equations for a particular $K > 0$. It turns out that even without knowing $K$, the unique solution to this equation system is the symmetric $p_1 = p_2 = p_3$. To prove this, assume that the three are not the same. WLOG we can assume that $p_1$ is among the maxima of $\{p_1, p_2, p_3\}$. Then we can distinguish two cases: First, imagine that $p_1 \geq p_2 \geq p_3$, where at least one of the two inequalities is strict. Then because $K > 0$, it is $p_1 + Kp_2 > p_2 + Kp_3$, contradicting the first equality in line 31. Second, imagine that $p_1 \geq p_3 \geq p_2$, where at least one of the inequalities is strict. Then it

---

[3]For example, in Mathematica, the following code identifies the unique solution $(1/3, 1/3, 1/3)$:
```
Solve[(4*13^3) * p1 * ((p1-1/4)*(p2-1/4)*(p3-1/4)) + p2 == (4*13^3) * p2 *
((p1-1/4)*(p2-1/4)*(p3-1/4)) +p3 == (4*13^3) * p3 * ((p1-1/4) * (p2-1/4)*(p3-1/4)) +
p1 && p1+p2+p3==1 && p1>=1/4 && p2>=1/4 && p3>=1/4, p1,p2,p3]
```

is $p_2 + Kp_3 < p_3 + Kp_1$, contradicting the second equality in line 31. In conclusion, it must be $p_1 = p_2 = p_3$ as claimed.

Now that we have shown that $(1/3, 1/3, 1/3)$ is the only strongly ratifiable policy, we can conclude by Theorem 2, that $\pi_t$ almost surely does not converge to any policy other than $(1/3, 1/3, 1/3)$. It now only remains to show that $\pi_t$ almost surely does not converge to $(1/3, 1/3, 1/3)$.

To show that $\pi_t$ cannot converge to $(1/3, 1/3, 1/3)$, we will analyze the history of what we will call *relative (empirical) Q-values*, which we will denote by $D_t(a_j, a_i) = Q_t(a_j) - Q_t(a_i)$. In order to converge to $(1/3, 1/3, 1/3)$, the relative Q-values must all converge to 0. In particular, it has to be

$$X_t := \sum_{a_i, a_j : i < j} |D_t(a_j, a_i)| \to 0, \tag{33}$$

as $t \to \infty$.

We will show, however, that these values almost surely do not converge to 0 if the policies converge to $(1/3, 1/3, 1/3)$. Roughly, we show that when the relative $Q$-values are close to 0 and the agent acts according to a policy that is close to $(1/3, 1/3, 1/3)$, the $Q$-values will in expectation be updated toward the action that is currently most likely to be taken. Thus for large enough $t$, $X_t$ will always increase in expectation. With some other easy-to-verify properties of $X_t$, we can then apply Lemma 13, which gives us that almost surely the $X_t$ do not converge to 0 as $t \to \infty$.

In order to prove that $\mathbb{E}[X_t \mid \mathcal{F}_{t-1}] - X_{t-1} > 0$ for large enough $t$ and assuming $X_t$ is close to 0 and $\pi_t$ close to $(1/3, 1/3, 1/3)$, let $a^* \in \arg\max_a \pi_t(a)$. Because of stochasticity of the rewards and by line 2, it is $\pi_t(a^*) > 1/3$ for large enough $t$. Further, let $a^- \in \arg\min_a \pi_t(a)$. It is $\pi_t(a^-) \leq 1/3$. Finally, let $\epsilon = \pi_t(a^*) - \pi_t(a^-)$.

The $X_t - X_{t-1}$ can be seen as the sum of three differences $|D_t(a_j, a_i)| - |D_{t-1}(a_j, a_i)|$. We start with the difference for $a^*$ and $a^-$. It is

$$\begin{aligned}
&\mathbb{E}\left[|D_t(a^*, a^-)| \mid \mathcal{F}_{t-1}\right] - |D_{t-1}(a^*, a^-)| \\
&= \alpha_t \left(\mathbb{E}[R(a^*, \pi_t)] - \mathbb{E}[R(a^-, \pi_t)]\right) - \alpha_t \left(Q_{t-1}(a^*) - Q_{t-1}(a^-)\right)
\end{aligned} \tag{34}$$

Now, assuming that $\pi$ is close enough to $(1/3, 1/3, 1/3)$ that $\pi(a_j) \geq 1/4 + 1/13$ for all $j$, it is

$$\mathbb{E}[R(a^*, \pi_t)] - \mathbb{E}[R(a^-, \pi_t)] \tag{35}$$

$$= (\pi(a^*) - \pi(a^-)) \cdot 4 \prod_j 13 \left(\pi(a_j) - \frac{1}{4}\right) + \pi(a^*_{+1}) - \pi(a^-_{+1}) \tag{36}$$

$$\geq 4\epsilon - \epsilon \tag{37}$$

It is left to estimate the other summands in the expectation of $X_t - X_{t-1}$. Consider any pair of actions $a_i, a_j$ with $i > j$. Because $|D_t(a_i, a_j)| = |D_t(a_j, a_i)|$, we can assume WLOG that $Q_{t-1}(a_i) > Q_{t-1}(a_j)$, which for large enough $t$ also means $\pi_t(a_i) > \pi_t(a_j)$. Thus, by similar reasoning as before,

$$\begin{aligned}
&\mathbb{E}[|D_t(a_i, a_j)| \mid \mathcal{F}_{t-1}] - |D_{t-1}(a_i, a_j)| \\
&= \alpha_t \left(\mathbb{E}[R(a_i, \pi_t)] - \mathbb{E}[R(a_j, \pi_t)]\right) - \alpha_t \left(Q_{t-1}(a_i) - Q_{t-1}(a_j)\right).
\end{aligned} \tag{38}$$

and

$$\mathbb{E}[R(a_i, \pi_t)] - \mathbb{E}[R(a_j, \pi_t)] \geq -\epsilon. \tag{39}$$

Thus, overall for large enough $t$ we have

$$\mathbb{E}[X_t \mid \mathcal{F}_t] - X_{t-1} \geq \alpha_t \epsilon - \alpha_t \left(\sum_{a_i, a_j : i < j} Q_{t-1}(a_i) - Q_{t-1}(a_j)\right) \tag{40}$$

By line 2, $\epsilon$ outgrows the differences in Q-values and therefore this term will be positive for all large enough $t$, as claimed. $\qquad\square$

# E  Proof of Theorem 7

**Theorem 7.** *Assume that there is some sequence of random variables $(\epsilon_t \geq 0)_t$ s.t. $\epsilon_t \underset{t \to \infty \text{ a.s.}}{\to} 0$ and for all $t \in \mathbb{N}$ it is*

$$\sum_{a^* \in \arg\max_a Q_t(a)} \pi_t(a^*) \geq 1 - \epsilon_t. \tag{3}$$

*Let $P_t^\Sigma \to p^\Sigma$ with positive probability as $t \to \infty$. Then across all actions $a \in \operatorname{supp}(p^\Sigma)$, $q_a(a)$ is constant.*

*Proof.* Consider any $a \in \operatorname{supp}(p^\Sigma)$ that is played with positive frequency. Because exploration goes to zero, almost all (i.e. frequency 1) of the time that $a$ is played must be from $\pi_t$ playing $a$ with probability close to 1. Therefore, whenever $P_t^\Sigma \underset{t \to \infty}{\to} p^\Sigma$ it is

$$Q_t(a) \underset{t \to \infty \text{ a.s.}}{\to} q_a(a). \tag{41}$$

Thus $q_a(a)$ must be constant across $a \in \operatorname{supp}(p^\Sigma)$, since otherwise the actions with lower values of $q_a(a)$ could not be taken in the limit. $\square$

# F  Proof of Theorem 8

**Theorem 8.** *Same assumptions as Theorem 7. If $|\operatorname{supp}(p^\Sigma)| > 1$ then for all $a \in \operatorname{supp}(p^\Sigma)$ there exists $a' \in A$ s.t. $q_a(a') \geq q_a(a)$.*

*Proof.* Let $|\operatorname{supp}(p^\Sigma)| > 1$ and suppose that $\exists a \in \operatorname{supp}(p^\Sigma)$ s.t.

$$\forall a' \in A - \{a\} : q_a(a') < q_a(a). \tag{42}$$

Policies close to $\pi_a$ are almost surely played infinitely often. Every time $T$ this happens we have that $Q_T(a) \geq Q_T(a')$ for all $a' \in A - \{a\}$. Now it is easy to see that if 42 holds, then there is a $K$ s.t. every such time $T$, there is a chance of at least $K$ that for all $t \geq T$ it is $Q_t(a) > Q_t(a')$ for all $a' \in A - \{a\}$. Hence almost surely $\operatorname{supp}(p^\Sigma) = \{a\}$, which contradicts the assumption that $|\operatorname{supp}(p^\Sigma)| > 1$. $\square$

# G  Proof of Theorem 9

**Theorem 9.** *Same assumptions as Theorem 7. Let $U$ be the Q-value $q_a(a)$ which (by Theorem 7) is constant across $a \in \operatorname{supp}(p^\Sigma)$. For any $a' \in A - \operatorname{supp}(p^\Sigma)$ that is played infinitely often, let frequency 1 of the exploratory plays of $a'$ happen when playing a policy near elements of $\{\pi_a \mid a \in \operatorname{supp}(p^\Sigma)\}$. Then either there exists $a \in \operatorname{supp}(p^\Sigma)$ such that $q_a(a') \leq U$; or $q_{a'}(a') < U$.*

*Proof.* Suppose there is an $a' \in A - \operatorname{supp}(p^\Sigma)$ for which both are false, i.e. $q_a(a') > U$ for all $a \in \operatorname{supp}(p^\Sigma)$, and $q_{a'}(a') \geq U$. Frequency 1 of the time that $a'$ is played is when the policy is near an element of $\{\pi_a \mid a \in \operatorname{supp}(p^\Sigma) \cup \{a'\}\}$, and so $Q_t(a')$ converges to some convex combination of $q_a(a')$ for $a \in \operatorname{supp}(p^\Sigma) \cup \{a'\}$. Therefore, in the limit $Q_t(a')$ is bigger than $U$. But that is inconsistent with $a'$ being played with frequency 0. $\square$