# OpenReview forum: "Reinforcement Learning in Newcomblike Environments"
_NeurIPS.cc/2021/Conference — NeurIPS 2021 Spotlight_

### Official Review · Reviewer_4LMa · 2021-06-25

**Rating:** 7
**Confidence:** 3

**Summary:**

The paper formalizes Newcomblike decision processes (NDPs), a generalization of Markov decision processes in which reward and transitions may depend on the agent's policy. This formalism better reflects how e.g. self-driving car policies (aggressive vs meek) affect how humans drive (back off vs take advantage of the docile AI). The authors prove a range of theoretical results about when model-free value-based RL algorithms converge to a policy, what that policy looks like, etc. The authors run simple experiments to validate their claims.

**Limitations And Societal Impact:**

The authors clearly discuss the scope of their theoretical and empirical contributions. They note conjectures for future work.

Societally, this work seems beneficial, and I'd like to see the authors discuss the benefits of better understanding Newcomblike problems with respect to e.g. self-driving car policies.

**Main Review:**

**Originality.** This work formalizes several key concepts from the decision theory literature and situates them in the RL context. There is little past work in this setting, which makes this contribution especially valuable IMO.

**Quality.** The work seems sound, although I haven't gone through most of the math. The experiments are simple but illustrative.

**Clarity.** I overall found the paper clear. The authors use intelligent examples and reasoning throughout the paper, and the graphics are good.

That said, I have several points of confusion:
* Below page 89, what does "w.p." mean? Given that you have extra room, can you remove the abbreviation?
* I didn't understand 304-313 explaining theorem 9. Maybe a more concrete example would help.

**Significance.** I think this paper is important, especially given the future work I hope it motivates. Newcomblike problems are understudied in the RL context, even though they are clearly relevant. The theoretical results are good, the experiments sensible, and the paper is well-written.

**Post-rebuttal**: I thank the authors for their clarifications. My score remains unchanged: accept.

**Time Spent Reviewing:**

4

---

> ### Author Response · Authors · 2021-08-11
> **Response to Reviewer 4LMa**
>
> We thank the reviewer for helping us improve our paper!
>
> >Below page 89, what does "w.p." mean? Given that you have extra room, can you remove the abbreviation?
>
> We will replace the abbreviation with “with probability” (here and in the definition of Death in Damascus).
>
> > I didn't understand 304-313 explaining theorem 9. Maybe a more concrete example would help.
>
> We see how this bit of text is relatively hard to understand. (Reviewer MhxB seems to agree as well.) Indeed, a concrete example would help. What makes this difficult is that we don’t have a game to illustrate, for instance, the mechanism described in 306-308. (We don’t have a game with actions $a$ and $b$ such that $a$ is generally good and $b$ is generally bad but $a$ is bad when the agent plays $b$ with high probability.) Introducing an entire new example might not be worth it (especially given that it would also take the reader some time and effort to understand the example). In any case, we will extend and rework this bit of text to make it easier to understand.
>
> >Societally, this work seems beneficial, and I'd like to see the authors discuss the benefits of better understanding Newcomblike problems with respect to e.g. self-driving car policies.
>
> We will add a short discussion of societal significance; see our general response.
>
> We again thank 4LMa for their comments and apologize for the delay.

---

### Official Review · Reviewer_MhxB · 2021-07-02

**Rating:** 7
**Confidence:** 3

**Summary:**

This paper introduces Newcomblike decision processes (NDPs), a generalization of MDPs where the transition probabilities depend on the agent's policy, representing scenarios like Newcomb's problem. The authors use an autonomous driving example to illustrate how Newcomblike dynamics may arise in the real world. They define ratifiability for policies and show that a value-based RL agent cannot converge to a non-ratifiable policy.  They introduce some examples (Repellor Problem and LARPS) of settings where this implies that the algorithm cannot converge to any policy, and investigate the limit behavior of non-converging algorithms (convergence to a specific action frequency).


**Limitations And Societal Impact:**

The paper adequately addresses limitations of this work but does not discuss possible societal impacts. It could be useful to discuss how better understanding of Newcomblike dynamics could potentially contribute to misuse of AI, e.g. in the example of autonomous vehicles discussed in the paper.


**Main Review:**

This paper addresses an important setting where the outcomes for the agent depend not only on its actions but on its entire policy. It shows how this setting can arise not only in thought experiments but also in the real world using the example of autonomous driving, and introduces a generalization of MDPs for this setting. The non-convergence results are a novel and significant theoretical contribution. I have skimmed the main proofs and they appear to be correct.

The main weakness of this paper that limits its impact is lack of clarity, which could be improved as follows:
- State the motivation more clearly in the introduction section, e.g. by moving the second paragraph of the discussion section to the introduction. Emphasize a take-home message for the readers, e.g. the implications of non-convergence and how you expect these results will contribute to "developing extensions of value-based RL that can detect and correct these failures".
- Define ratifiability in the abstract (which should be self-contained), and provide a reference for the standard definition of ratifiability referred to in section 2 as "what decision theorists mean when they speak about ratifiability".
- Use more readable formatting for the second-last paragraph of the introduction (e.g. a numbered list).
- Provide a statement of Newcomb's problem in the introduction, since not all readers will be familiar with the problem.
- Provide a plain language description of the Death in Damascus problem in section 1.3.
- Describe the theorems in section 4 (convergence of action frequencies) in plain language, since it is not obvious what they mean from the formal statements.
- Clarify how this paper differs from the previous work on Newcomblike problems referred to in "Another more sophisticated setting was studied by Oesterheld (2019)"


**Time Spent Reviewing:**

2.5

---

> ### Author Response · Authors · 2021-08-11
> **Response to Reviewer MhxB**
>
> We thank MhxB for their detailed suggestions on making the paper easier to understand. We will implement these recommendations as suggested.
>
> >Clarify how this paper differs from the previous work on Newcomblike problems referred to in "Another more sophisticated setting was studied by Oesterheld (2019)"
>
> We don't want to give too much space to discussing that paper, because it is tangential and discussing it in detail may give the false impression that it is particularly closely related. Nevertheless, we plan to replace the sentence in the paper with something like the following. "Oesterheld (2019) studies a setting where agents make decisions in Newcomblike environments with the goal of maximizing a reward provided by an overseer. However, he does not consider the learning process. Instead he assumes that the agent has already formed beliefs and uses some form of causal or evidential decision theory. He also considers the case where the overseer rewards based on beliefs, as opposed to having the reward come directly from the environment.”
>
> We will add a short discussion of societal significance; see our general response.
>
> We again thank the reviewer for their contributions and apologize for the delay in responding.

---

> > ### Comment · Reviewer_MhxB · 2021-08-14
> > **Thanks**
> >
> > Thanks for the response! With the improvements to the clarity of the paper, I am happy to increase my score.

---

### Official Review · Reviewer_L2sm · 2021-07-17

**Rating:** 7
**Confidence:** 4

**Summary:**

This paper introduces Newcomblike Decision Problems (NDPs), an analog of MDPs in which the environment (specifically the transitions and reward) may depend on the agent’s _policy_ in addition to the state and action. They theoretically study the performance of standard RL algorithms when applied as is to continuous NDPs. Key results include:

1. Any RL algorithm that eventually finds accurate Q-values for its policy, and is greedy with respect to those Q-values, will converge to a strongly ratifiable policy, or will not converge at all. A strongly ratifiable policy is one that never “wishes” it could switch its action while keeping its “identity” the same. (The informal description is mine; the paper presents a formal statement.) SARSA, Expected SARSA, and Q-learning all satisfy the conditions for this theorem.
2. In any continuous NDP there exists a “strongly ratifiable” policy, but it may not be the optimal policy (for example, in the classic Newcomb problem).
3. Nonetheless, there exist continuous NDPs where typical RL algorithms almost surely do not converge to the strongly ratifiable policy.
4. When there is not infinite exploration of states and actions, RL algorithms instead satisfy weak ratifiability, where the policy never “wishes” it could switch its action to some other action that is explored infinitely often (but may “wish” to switch actions to an action that isn’t explored infinitely often).
5. In the special case of bandit NDPs, similar results can be proven for algorithms with limiting _action frequencies_ (even if there is no limiting policy).

**Ethical Concerns:**

No ethical concerns.

**Limitations And Societal Impact:**

See main review.

**Main Review:**

Overall score
-------------

**Originality:** While there has been some prior work on testing RL algorithms in Newcomblike problems (which the authors mention), to my knowledge there has not been any theoretical analysis of the kind done in this paper.

**Significance:** Newcomblike problems seem like an interesting problem to study: RL agents will be deployed in environments that then respond to the deployment of the RL agent (as in the autonomous driving example). While I tend to think of such situations from the lens of game theory, it can also be thought of as a Newcomblike situation. (An example of a paper studying a specific practical problem of this sort is Milli et al, “The Social Cost of Strategic Classification”, and various papers on recommender systems.) This paper provides some basic results for NDPs, establishing the basic theory on which future papers can build; this seems quite worthwhile.

The main argument against significance is that the paper studies RL algorithms applied to NDPs, even though RL algorithms are designed to work with MDPs and are not expected to work well on NDPs. Nonetheless, I agree with the authors that in practice it seems likely that we will apply RL algorithms to NDPs, and so it is worth studying this setting (for example, OpenAI Five was trained with PPO, but is effectively playing in an NDP since its human opponents know how it was trained and can adapt to its policy).

**Quality:** The results are interesting and look correct to me (though I have not checked the proofs in detail).

**Clarity:** I found the paper clear and surprisingly easy to read given the significant number of theorems. I have mentioned some nitpicks at the end of this review.

Relation to game theory
-----------------------

Newcomblike problems are often meant to capture cases in which the environment is “bigger” than the agent, i.e. the agent is unable to model the environment in perfect detail. This also happens in multiagent settings / game theory, since the agent cannot have a perfect model of other agents (e.g. if another agent is a copy of the agent, then the agent would have to have a perfect model of itself, which tends to lead to contradiction).

The authors briefly mention game theory in the related work, but it seems like there is much more to be said here. Can we reduce NDPs to game theory? In particular, it seems like we could take an arbitrary NDP and reduce it to a Stackelberg game (leader-follower game) in which the original agent is the leader and the environment is the follower. Does this reduction provide insight?

Many of the motivating examples for NDPs can also be modeled as games in which you are searching for Nash equilibria, e.g. in autonomous driving you want to find a policy that performs well _given how human drivers will respond_ -- this is very similar to what a Nash equilibrium would give you (to the extent that iterated best response in the “autonomous driving” game converges to a Nash equilibrium). As a result, the topic of this paper is very related to the large literature on multiagent RL, much of which is concerned about whether typical RL algorithms converge to Nash equilibria (just as this paper considers whether typical RL algorithms converge to optimal or strongly ratifiable policies). For example (the first is most relevant):

Mazumdar, Eric, Lillian J. Ratliff, and S. Shankar Sastry. "On gradient-based learning in continuous games."
Letcher, Alistair, et al. "Stable opponent shaping in differentiable games."

Questions
---------

How do NDPs relate to game theory and multiagent RL? What novelty do NDPs introduce that could not already be modeled in these areas?

I see the point of strong ratifiability: a strongly ratifiable policy never “wants to” switch to a different action (in this respect it is similar to a Nash equilibrium in game theory). What’s the point of weak ratifiability? My best guess is that in a weakly ratifiable policy, the policy never “wants to” switch between actions within its support -- but wouldn’t it “want to” switch to an action not in its support? Why do we care only about actions in the support of the policy? (Maybe the point is just “that’s what’s needed to make Theorem 5 work”, since it could be the case that the policy should “want to” switch to an action that hasn’t been infinitely explored.)

Minor nitpicks
--------------

Since you are introducing a new formalism, please define what an optimal policy is in this new formalism (I assume it is any policy in the set $\arg \max_{\pi} \mathbb{E} [ \sum_{t=0}^{\infty} \gamma^t R(s_t, a_t, s_{t+1}, \pi) ]$, which you also write as $\arg \max_{\pi} \mathbb{E}[R \mid \pi]$).

> The key feature of this NDP is that regardless of the policy a2 (“two-boxing”) yields a higher reward than a1 (“one-boxing”).

I wouldn’t say this is _regardless_ of the policy, as the policy very much does influence the reward. I might instead say something like “The key feature of this NDP is that for any fixed policy, a2 (“two-boxing”) would yield a higher reward than a1 (“one-boxing”).”

> This is because the optimal policy in Newcomb’s Problem does not select optimal actions.

I’m not sure what this means, and it doesn’t seem like an explanation. I would say something like “this is because the choice of which policy is optimal depends not only on actions, but also on the policy itself; the Bellman optimality condition does not include the effect of the policy on optimality”.

> it is clear that all actions in the support of π∞ must have equal expected utility given π∞.

I would change this to “it is clear that, _for a given state_, all actions in…”

**Time Spent Reviewing:**

3

---

> ### Author Response · Authors · 2021-08-11
> **Response to Reviewer L2sm**
>
> We would like to thank Reviewer L2sm for their feedback!
>
> We would first like to emphasize that NDPs cannot be reduced to games in general -- we have discussed this in more detail elsewhere in our response. Regarding Stackelberg games in particular: We agree that these are in some ways more similar to our model than simultaneous games, since we consider the choice of a single agent. Interestingly, both the agent and the environment can be seen as the Stackelberg leader/follower. In a way, our environments “move first” by specifying some dynamic and the agent learns to choose in response to the given dynamic. (For example, we could consider the NDP (or even MDP) of playing the Game of Chicken against a player who always dares.) But if we imagine that our NDP best responds to the agent’s policy, we can view the agent as the Stackelberg leader and the NDP as the follower. In any case, NDPs still differ from Stackelberg models in the ways outlined in our general response. In particular, the environment need not react to the agent’s chosen policy in an expected-utility maximizing way.
>
> As for the point of weak ratifiability -- the reason that we define and discuss this notion is not that we believe this to be a good notion of optimality, but rather that we believe that it could be a useful notion for describing the limit behaviour of certain kinds of learning algorithms, in particular those that explore some actions only finitely many times (e.g., ones that have empirically resulted in bodily harm). In principle, one could also give more fine-grained, quantitative results related to weak ratifiability. For example, a typical question in statistical learning theory might be: If the agent tests each action at least 100 times, what is the probability that it will end up with roughly (up to $\epsilon$) accurate empirical estimates of the true $Q$ values?
>
> Regarding the minor nitpicks -- we very much thank the reviewer for pointing these out! We will incorporate them in the final version of the text.
>
> Regarding Mazumdar et al.: We thank the reviewer for bringing up this paper. This work seems similar to ours in that it studies a fairly generic type of learning algorithm. The algorithm is substantially different from ours in that it is directly based on gradients of the choice variable. (In our case, that would be the gradient of the policy.) We will add a discussion of this work in the “Learning in Games” subsection of the related work section.
>
> Regarding Letcher et al.: We should discuss this as well in the related work subsection on MARL. We find this line of work very interesting and probably there is some overlap in readership. But it does seem to be quite different from our paper in some ways. For one, SOS specifically addresses (among other issues) the problem of giving a learning algorithm that works well when playing against itself, or other learning algorithms. Whereas, we study a setting in which a single agent has to adapt to a fixed, given environment. To enable opponent shaping, Letcher et al. assume that the agent knows the game’s payoff matrix including the other agents’ utility functions. Our Q-learning starts learning “from scratch”; and we don’t assume that the environment contains agents that (learn to) maximize some expected utility. We will also write a more general comment on the relation of our work to MARL.
>
> We thank L2sm for their careful review and apologize for sending our response late.

---

> > ### Comment · Reviewer_L2sm · 2021-08-12
> > **Thanks**
> >
> > Thanks for the response! I broadly agree with it.
> >
> > For the reduction to a Stackelberg game, I was imagining that in the Stackelberg game the agent would choose a _policy_ instead of an action, and the environment would respond to the choice of policy. That is, the rows of the payoff matrix would be indexed by policies rather than by particular actions. (I think it's plausible that this reduction is so "general" that there is no insight to be gained from it.)

---

> > > ### Author Response · Authors · 2021-08-14
> > > **Thoughts on Stackelberg**
> > >
> > > Thanks for elaborating! Generally we think the reviewer is right that the reduction requires a too general version of Stackelberg models to be insightful.
> > >
> > > From our understanding, having the agent choose a policy and letting the environment/Stackelberg follower best respond to the policy is within the usual kind of Stackelberg models that people consider. (Some of the earliest game theory works (like the Minimax theorem) are on playing zero-sum games sequentially in exactly this way.)
> > >
> > > However, the Stackelberg follower would usually still maximize the expectation of some utility function given the leader’s chosen probability distribution. For example, let’s say the leader has actions $a$ and $b$ and the follower has actions $c$ and $d$. Then the leader’s policy is simply a probability $p_a$ of playing $a$. The follower would then choose the action (or one of the actions) $x$ that maximize(s) $p_a u_{\mathrm{follower}}(a,x) + (1-p_a) u_{\mathrm{follower}}(b,x)$ over $x= c,d $. This strongly limits the kinds of ways in which the environment can respond. Generally there will be some threshold $p_a^*$ at which the follower is indifferent (and so might mix in arbitrary ways). Above this threshold the follower deterministically plays one action, and below this threshold the follower deterministically plays the other action. The environments in our versions of Death in Damascus and Newcomb’s problem cannot be modeled as such threshold functions.
> > >
> > > There are ways to fix this, but we think these take us outside the realm of the kind of Stackelberg models that people usually consider. For instance, one could let the follower assign values directly to pairs of probability distributions. (E.g., if the Stackelberg follower has a utility function $u(p_a,p_c)=-|p_a-p_c|$, it will simply respond to any $p_a$ by playing $p_c=p_a$. But we can trivially model any object/environment as maximizing expected utility if we can specify that it assigns high utility to reacting in whatever way it reacts.)
> > >
> > > Of course, even if we considered a version of Newcomb’s problem in which the predictor plays a best response in some sense, then typical Stackelberg-type work wouldn’t consider the challenge posed to learning agents by the policy/action interaction.

---

> > > > ### Comment · Reviewer_L2sm · 2021-08-15
> > > > **Makes sense**
> > > >
> > > > Yeah, the requirement that the follower maximize expected utility does in fact restrict potential environments. Thanks!

---

### Official Review · Reviewer_xfAj · 2021-07-29

**Rating:** 7
**Confidence:** 3

**Summary:**

The paper begins by defining a Newcomblike decision process (NDP), a natural extension of MDP in which both the reward and/or dynamics are allowed to depend on the learner's policy. The main theoretical claims of the paper are largely three-fold:
1. define two notions of ratifiability (strong and weak). strong ratifiability can be seen as a policy that always selects actions among those that are optimal w.r.t. the q-values of the policy, while weak can be seen as q-values for a policy being constant among those actions sampled by the policy
2. show that q-values, when converge, must converge to a strong ratifiable policy; this can relaxed to weak ratifiability if q-values are accurate only for state-action pairs sampled infinitely often
3. with the help of several bandit examples, show that standard RL such as QL and SARSA is not required to converge at all.

**Limitations And Societal Impact:**

===== possible limitations and/or questions for authors: =====

significance: The definition of "environment" in this paper is already quite different than what standard RL is able to solve, single they are tailored for "stationary" problems, whereas here the environment responds to the agent's policy in ad-hoc ways. Based on this, it is not clearly motivated why standard RL like Q-learning and SARSA would be natural starting points for solving the NDPs. Furthermore, is the NDP the best way to model the AV example in the intro? Wouldn't a standard game-theory formulation be able to deal with such problems, where the agent (AV) plays adversarially against nature (a single human driver or set of drivers)? Therefore, wouldn't an equilibrium-based strategy, such as Nash-Q, fictitious play, or other game-theoretic RL method do better for solving problems, given the vast literature already available for solving them? alternatively, could a restricted class of NDPs with certain structure (like a budget on how much the environment allowed to vary w.r.t policies) be suitable for standard RL with some sort of weak guarantees?

connection to multi-agent RL: In relation to above commentary, I do believe more discussion should be provided to shed light on how the NDP + RL formulation is more meaningful for solving the kind of problem motivated early on than typical game theory, or what kind of problems can be solved with the new formulation that cannot otherwise. It almost seems that the NDP formulation is a balance between single agent and multi agent RL, where we only learn from direct environment responses to our behaviors (which need not even be optimal from the nature's point of view), rather than not at all or trying to fully account for natures' best response to our behavior. the example provided shows that human drivers could directly formulate a best response to the AV policy. In light of this, i believe more needs to be discussed on this matter and make the connection with other game-theoretic notions more clear for readers.

other suitable formulations within ordinary RL: Since the environment here is policy conditioned, wouldn't it be possible to embed the policy conditioning within standard RL frameworks such as Q-learning? Such work has already been considered in the context of policy evaluation, see. e.g. Harb et al, 2020, or Borsa et al., 2018 for two examples in the RL context. If the Q(s, a; pi) is policy conditioned and standard RL with Bellman principle applied, then don't the problems with instability claimed here in sec. 4 disappear altogether, since now Q can assign credit appropriately in a way consistent with the true environment?

===== some other minor comments: =====

- would be nice to label axis in graphs, e.g. label x axis as \alpha in Fig 2a and "Probability of Convergence" as y, etc. I would make the axis numbering larger to make it easy to read as the font is too small.
- the idea of Newcomblike decision process does not appear in the cited papers apart from the single stage problem, as far as I can see. It would be nice to cite the appropriate reference to this modeling framework if it is previously considered in Sec 1.1.

===== references =====

Harb, Jean, et al. "Policy evaluation networks." arXiv preprint arXiv:2002.11833 (2020).
Borsa, Diana, et al. "Universal successor features approximators." arXiv preprint arXiv:1812.07626 (2018).

**Main Review:**

Overall, I believe this paper is quite novel from a conceptual standpoint, presenting the idea of Newcomblike process for making decisions in long-term horizons. This formulation seems quite intuitively appealing, as do the paper's main contributions showing that either standard RL converges to a suitable notion of "optimality" or not at all (which can be viewed as an example of 0-1 law). I agree with the authors that a negative result here, namely the possible failure modes of standard RL algorithms, can be a good starting point for designing ones that do, though it is not fully clear how the analysis in the paper claims to be able to proceed from here. However, I believe the results could be important for shaping future work and have some impact in game-theoretic work in RL in the future.

My main concerns, which I discuss in the limitations section, is that it becomes apparent fairly early that standard RL algorithms are not suitable here due to the policy dependence of the environment. I do not have any problem with providing negative results, but only if their outcome is fairly significant and insightful and can provide clear directions how to proceed from here and shape future algorithms, which I am not entirely sure here. As I read the paper I was wondering early on whether a game theoretic formulation of the agent could have made more sense as a starting point right away, rather than suggesting standard single agent QL (e.g. Nash-Q, fictitious play etc). I also have some slight issues with clarity of figures which I discuss later.

**Time Spent Reviewing:**

2

---

> ### Author Response · Authors · 2021-08-11
> **Response to Reviewer xfAj**
>
> We would like to thank Reviewer xfAj for their feedback!
>
> First, we agree that it fairly quickly becomes apparent that standard model-free RL algorithms will not be suitable for solving NDPs in general. However, we would like to note that we go beyond this by analyzing _how_ they fail, _when_ they will fail, and _why_, in a way that is quite nuanced and comprehensive. Therefore, this work could (partly) be seen as a type of robustness analysis, that explores how standard RL algorithms will behave in certain kinds of edge cases and under certain kinds of model misspecification. Of course, we also hope that this analysis will provide useful tools for designing new algorithms as well.
>
> For a discussion of the relation of our work to game theory, see the general comment. In short, there are decision problems that can be modeled as NDPs but not as games between expected utility maximizers (and vice versa). Moreover, we think it is often fruitful to analyze a problem from many different angles and perspectives, and while there has already been much work analyzing the behavior of RL agents through a "game-theoretic lens", this is the first work that analyzes them through a "decision-theoretic lens".
>
> We agree that it could be fruitful to study restricted classes of NDPs to obtain weak guarantees about the behaviour of standard RL algorithms in these cases. We like the reviewer’s idea of assuming limited budgets on how or how much the environment can depend on the policies. Another interesting case might be Death in Damascus/Game of Chicken-like dynamics, where the strongly ratifiable policy is optimal. We consider these to be exciting directions for follow-up work and will add them to the "Discussion and Further Work" section!
>
> Since multi-agent RL has also been brought up by L2sm (also in a reply to our earlier general response), we will discuss this in a “general” comment.
>
> We would also like to thank the reviewer for bringing the works by Harb et al and Borsa et al. to our attention! We were not aware of this existing work on learning to assign values directly to policies. We will add this in our future work section as well. It would be very interesting to see how much modification would be required to get Harb et al.’s approach to work efficiently in NDPs.
>
> We will implement the suggestions on axis labeling.
>
> We will add a short note on discussions of multi-stage Newcomblike decision processes in prior work. Indeed, most prior work only discusses single-stage decision problems. We believe our formal model of sequential Newcomblike decision problems to be novel. The only other general model of sequential Newcomblike problems that we are aware of is due to Everitt et al. (2015): Sequential Extensions of Causal and Evidential Decision Theory. https://arxiv.org/abs/1506.07359 However, their model is quite different from ours. It does not consider mixed policies, for example. Of course, it is also quite common for decision theorists to discuss specific sequential Newcomb-like problems without a general model. For example, see Arif Ahmed’s draft on “Sequential Choice and the Agent's Perspective” https://www.academia.edu/36270656/Sequential_Choice_and_the_Agents_Perspective or Section III of Oesterheld and Conitzer’s (2021) “Extracting Money from Causal Decision Theorists” https://academic.oup.com/pq/advance-article/doi/10.1093/pq/pqaa086/6118001
>
> We again thank the reviewer for their efforts and apologies for the delay in sending this response.

---

> > ### Comment · Reviewer_xfAj · 2021-08-24
> > **Thanks for addressing my concerns**
> >
> > Thank you for your detailed responses, and apologies on my end for replying late. Overall, I couldn't agree more that it is critical to understand why well-studied algorithms fail, and that studying the problem from a new perspective is also equally useful. Given my (now hopefully better) understanding on the distinction between games and NDPs, this paper's results seem even more valuable.
> >
> > As a general comment/suggestion, given how relatively poorly-studied NDPs have been as compared to MARL and that this question has been raised in other reviews, I am wondering if it could benefit the paper to allude to this distinction early on in the intro, making an even stronger point that -- as this reduction is not always possible -- standard RL is not a viable solution and demands further research to develop the *right* methods to solve it. I feel that the related work section could clarify a lot of concerns readers may have reading the paper's intro. Particularly, as one could interpret a human driver to be formulating a "minimax/best response", which I can see better now is not required  according to the very general definition of NDP in this work. In any case, I am happy to raise my score, thanks.

---

> > > ### Author Response · Authors · 2021-08-26
> > > **Thanks for the suggestion**
> > >
> > > Thanks a lot for this suggestion! We agree that this would be helpful. We will add a brief discussion to the introduction, with a forward reference to the planned section on related MARL work.

---

### Author Response · Authors · 2021-08-11
**General response**

We would like to thank all four reviewers for their detailed comments and helpful suggestions! In this comment, we discuss two points that multiple reviewers have at least touched on: our work’s societal impact and relation to game theory. We will discuss further points in comments to the individual reviews, which we will post tomorrow (August 11). We apologize for the delay in submitting our responses.

First, regarding societal impact: We agree with the reviewers that we should discuss this in more detail in the camera-ready version. In the following two paragraphs we briefly outline our views on this, which we will also (perhaps in somewhat shorter form) add to the paper.

We believe that our approach will enable more accurate reasoning about the behaviour of RL agents in more kinds of real-world situations, especially when interacting with humans or other agents. It is therefore a useful tool for assessing to what extent such agents will behave as we intend in Newcomb-like situations. We hold such improvements in understanding to be broadly beneficial to the safe design and deployment of AI systems.

Besides improving theoretical understanding, we hope that our work will eventually contribute to the design of AI systems that deal more successfully with real-world Newcomb-like dynamics. We generally believe such improvements will increase the safety of AI systems. For example, in the autonomous vehicle case it seems important for the system to figure out that following a simple or human-interpretable policy confers safety benefits. However, since many real-world Newcomb-like situations involve interactions with humans with conflicting goals, improvements should be analysed and considered carefully, especially if they increase an AI system’s ability to “win” conflicts to the detriment of human players. For instance, consider Newcomb’s problem as a simplistic model of an interaction between an autonomous vehicle and a human driver (as briefly discussed in the paper). Then we can interpret our results as saying that per default value-based RL learns to bow to a human bully’s will (i.e., it learns to let the human driver cut the AV off). In the case of autonomous vehicles, this appears to be an obstacle to real-world deployment. We therefore expect future autonomous vehicles to be designed to resist such bullying, to the bullies’ chagrin. While such a development is probably fine (even necessary) in this specific scenario, in other application scenarios one may be more inclined to let humans keep the upper hand.

Second, regarding the relation to game theory: Again we agree that we should dedicate more space in the paper to discussing this. Indeed, games have been on our minds throughout working on this project, as can be seen from our use of games as examples and the discussion of learning in games in Sect. 5.1. We hope that our work contributes to understanding dynamics of game-theoretic interactions specifically.

That said, in general there seem to be substantial differences between our NDPs and games. NDPs are more general in that the environment can react arbitrarily to the agent’s policy -- the environment need not play a best response to the agent’s policy. For instance, in Newcomb’s problem, the environment simply matches the agent’s policy, whereas best response functions are step functions in general. We expect that in many real-world situations, other agents in the environment (especially humans) cannot be comfortably modeled as traditionally rational (expected utility maximizing) agents. We think interactions with autonomous vehicles can serve as examples. Most people probably don’t reason rationally about small-probability, big-impact events, such as crashes involving autonomous vehicles. Also, humans will generally operate on a drastically simplified model of an AV’s policy (even when more detailed models are available). This is not to say that game-theoretic analyses of, for example, autonomous vehicles are not fruitful, but we believe that the NDP perspective can provide additional insight into learning in such situations.

At the same time, the NDP setting also makes a few assumptions to avoid the difficulties that game theory addresses. In game theory, we try to solve a problem from the perspective of multiple decision makers with potentially different objectives. In contrast, we take the perspective of one agent, taking the other agents/environment as given (without assuming their rationality). This makes it much easier to define, for example, what the optimal policy is. For most of the paper, we also assume that the environment is continuous in the agent's policy (while best response functions are usually discontinuous). By “assuming away” some of the problems of game theory, we can better focus on the Newcomb-like dynamics in learning, at the cost of having little to say about how two AI systems should learn to jointly play a game.

In a way, much of the above is analogous to the relation between the study of Newcomb-like problems in decision theory and the study of strategic interactions in game theory (independent of ML and AI). Probably most decision theorists who have studied Newcomb-like problems are also interested in game theory. (Many have studied the connection explicitly.) However they still consider it useful to study Newcomb-like problems that are not games (perhaps for reasons analogous to the above) or even to study games as Newcomb-like problems without game-theoretic tools (e.g., to provide a decision-theoretic foundation for game theory). Overall, the study of Newcomb-like decision problems overlaps with the study of strategic interactions in game theory, but neither subsumes the other.

We again thank the reviewers for their efforts! We are happy to continue the discussion on these issues and answer further questions if it is helpful.

---

> ### Comment · Reviewer_L2sm · 2021-08-11
> **Consider MARL as well**
>
> > NDPs are more general in that the environment can react arbitrarily to the agent’s policy -- the environment need not play a best response to the agent’s policy. [...] In game theory, we try to solve a problem from the perspective of multiple decision makers with potentially different objectives. In contrast, we take the perspective of one agent, taking the other agents/environment as given (without assuming their rationality). This makes it much easier to define, for example, what the optimal policy is.
>
> Indeed, this is why I think the most related work is in the multiagent reinforcement learning (MARL) literature, where the agent must learn to play in response to other agents whose policies are not known in full and may not be optimal for the specified utility function. (This is especially true in cases where we do decentralized training, in which case we really are "taking the perspective of one agent".)

---

> ### Author Response · Authors · 2021-08-13
> **On multi-agent reinforcement learning**
>
> Since multi-agent reinforcement learning (MARL) has also been brought up by two reviewers, we discuss it in this general thread, too. We plan to add a subsection on MARL in the related work section. However, we might benefit from further comments to make sure we aren’t missing anything (e.g., any further papers that might be particularly relevant to our work).
>
> As with game theory, we did have MARL in mind when writing and working on this paper. (In fact, some of us (the authors) have worked/are working on MARL ourselves.) However, we think that many of our comments on game theory apply in analogous ways to MARL as well.
>
> Relative to game theory, MARL drops the assumption that the opponent is ideally rational. But most MARL work still studies settings in which the opponent is learning to play best responses. In contrast, our NDPs may respond in ways that do not maximize the expectation of any utility function.
>
> Relatedly, much MARL work starts from the assumption that part of the environment is controlled by another agent. Usually the agents even start with knowledge of the other agents’ utility function. (More often than not, much stronger assumptions are made. For example, much MARL work only considers the zero-sum or only the fully cooperative case. See, e.g., [Zhang et al.’s recent overview paper](https://arxiv.org/pdf/1911.10635.pdf).) In contrast, we are concerned with learning completely “from scratch”, not knowing what type of dynamics the agent is facing. (In fact, we are partly motivated by scenarios in which the agent is explicitly designed under the false assumption that it will face no Newcomb-like dynamics.) There are exceptions to this in MARL, of course, such as the work by Mazumdar et al. cited by Reviewer L2sm, or early work by [Claus and Boutilier (1998)](https://www.aaai.org/Papers/AAAI/1998/AAAI98-106.pdf) . We should say that in many settings there are, of course, good reasons to make MARL-type assumptions: they are often at least approximately true and very productive.
>
> Also, almost all of the MARL work we are aware of asks how an agent should learn given that the other agents are also trying to learn. It’s also common to explicitly study what happens if multiple agents use a proposed learning algorithm. For instance, this is very much true of Letcher et al.’s (2020) work on stable opponent shaping (as well as some of its intellectual predecessors, such as [Foerster et al.’s (2018) LOLA](https://www.cs.cmu.edu/~mshediva/assets/pdf/lola-aamas18.pdf) and [Conitzer and Sandholm’s (2007) AWESOME](https://link.springer.com/content/pdf/10.1007/s10994-006-0143-1.pdf)). Our environments can be viewed as incorporating some non-stationarity relative to MDPs. But technically speaking we still study single-agent learning against a stationary environment. (If the agent uses some policy $\pi$ for a while at time step 1 and then again uses $\pi$ for a while at time step 1,000,000 we can expect the same result in NDPs but not in most MARL settings.)

---

### Decision · Program_Chairs · 2021-09-27

**Decision:**

Accept (Spotlight)

**Comment:**

The reviewers agree that the formalism introduced in the paper is both interesting and potentially useful. Newcomblike decision processes (NDPs) are a generalization of Markov decision processes (MDP). As such, they augment the space of environments we can model and can potentially lead to agents that are better equipped to tackle real-world problems. Since reinforcement learning algorithms used out-of-the-box do not always yield the most desirable behavior, the NDP formalism also invites future research on the design of new algorithms.

We encourage the authors to elaborate on the connections between NDPs and both game theory and multi-agent reinforcement learning, since this point has come up in multiple reviews and was one of the main themes of the discussion phase. A discussion on the societal impact of the proposed approach also seems particularly pertinent. Finally, we also ask the authors to carefully consider the reviewers’ suggestions to improve the presentation.